# Experimental Evaluation of the Performance of a Three-Phase Five-Level Cascaded H-Bridge Inverter by Means FPGA-Based Control Board for Grid Connected Applications

Fabio Viola

Dipartimento Energia, ingegneria dell'Informazione e modelli Matematici, DEIM, University of Palermo, 90133 Palermo, Italy; fabio.viola@unipa.it; Tel.: +39-091-238-60253

**Abstract:** Over the last decades, plants devoted to the generation of green energy significantly increased their number, together with the demand of same electrical energy, also stored in battery systems. This fact produced the growth of energy conversion systems with advanced performances with respect to the traditional ones. In this circumstance, multilevel converters play a significant role for their great advantages in performances, flexibility, fault-tolerability, employment of renewable energy sources and storage systems and finally yet importantly reduced filter requirements. In this context, this paper faces the performance of a cascaded H-bridge 5 level inverter in terms of harmonic distortion generated and injected into the grid. Through an accurate analysis that takes into account the pulse width modulation (PWM) multicarrier modulation techniques (phase disposition PD, phase opposition disposition POD, alternative phase opposition disposition APOD, phase shifted PS) and related reference signals (sinusoidal reference; third harmonic injection THI reference, switching frequency optimal SFO reference), a framework of distorting harmonics is presented by comparing twelve cases. The results obtained from the simulations are reproduced and validated in a prototype system of five level cascaded H-bridge multilevel inverter. A deep discussion of control and filtering system is provided to justify the choice of the best modulation technique to adopt.

**Keywords:** Cascaded H-bridge multilevel inverter (CHBMI); field-programmable gate array; total harmonic distortion (THD); modulation techniques

## 1. Introduction

The increased demand of green energy has led to the development of even more performing structures allowing the generation and storage of energy in DC form. The drawbacks are due to the increased number of harmonics introduced and filtered in the power grids.

The three main multilevel power inverter (MPI) structures proposed in technical literature, with their related benefits and disadvantages, according to [1–3] are diode clamped converter (DCC), capacitor-clamped inverter (CCI) and cascaded H-bridge (CHB) multilevel inverter.

The neutral point clamped converter (NPC) was the first multilevel structure proposed. The common DC-link is composed of four capacitors connected in series that split the voltage into four level. The middle point of the capacitors $n$ is used as neutral point. The peculiar components, that differentiate this circuit from the others multilevel inverters, are the clamping diodes that allow to subdivided the DC voltage on the switches. Thus, the voltage across on the switches is limited to one capacitor voltage equal to $V_{dc}/(n_L - 1)$, where $n_L$ is the number of level. By supposing that for each blocking diode its voltage value is identical to the voltage rating of active device, the number

of diodes requested for each phase will be $(n_L - 1) \cdot (n_L - 2)$. This converter presents some operative limits as: (1) max number of levels is five, due both to the complexity of the circuit and both to the large number of components demanded; (2) uneven distribution of semiconductor power losses among the switches, which reduces the switching frequency and the output power; (3) unbalanced capacitor voltages which generate low frequency harmonics; (4) the system cannot involve a modular structure (non-modular topology structure).

Flying capacitor inverters (FCIs) or CCIs are an alternative to overcome some of the DCC disadvantages. The structure of CCIs have similarity to NPC inverter except the CCI uses several capacitors in the place of the clamping diodes. The main advantage of the CCI are the redundant states to obtain the voltage levels. In this way, it is possible an even distribution of semiconductor losses among the switches but it is necessary a dedicated control algorithm to balance the capacitors voltage.

The increase of voltage levels confines the proper charging and discharging mechanism of capacitors. By considering the economical aspect, the cost of the inverter follows the increase of the number of levels, but also the device becomes bulkier and its lifetime decreases due to the growing number of used capacitors. For a $n_L$-level converter, it is necessary $(n_L - 1) \cdot (n_L - 2)/2$ clamping capacitors per phase in addition to the $(n_L - 1) \cdot$ main dc bus capacitors. Thus, the high number of capacitors limit the use to three or five levels. Moreover, lack of modularity and high quantity of capacitors for higher number of voltage levels reduces the reliability of this converter.

Figure 1 shows the topology structure of a single-phase five-level cascaded H-bridge multilevel inverter.

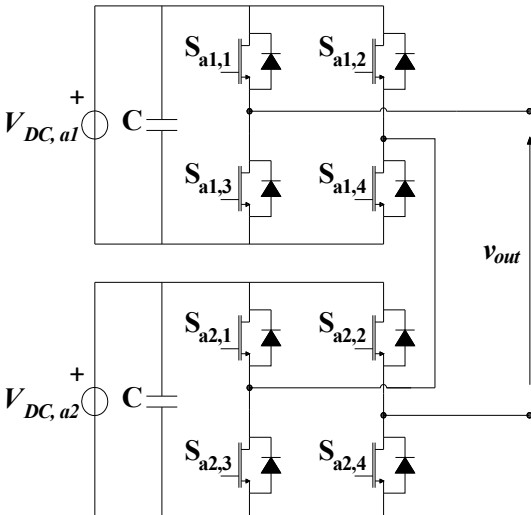

**Figure 1.** Topology structure of a single-phase five-level cascaded H-bridge (CHB) inverter.

This topology has a main advantage: the modular configuration, blocks can be added to reach voltage levels, control is easily performed, and maintenance, in case of fault, requires the disconnection of a block to keep the system working. Thus, each module can be either half- or full-bridge with separated DC source and can be controlled as a single-phase converter. This topology reaches in output medium voltage levels, by enforcing only common low-voltage components so there are not operative limits about max number of the voltage levels. Matching the number of capacitors and diodes between the cited topologies yields that CHB converter has the least number of components.

The phase voltage is synthesized by the addition of the voltages generated by the different modules. Thus, the voltage levels $n_L$ depend of the number of modules connected in series for phase through the equation:

$$n_L = 2n_{HB} + 1 \tag{1}$$

where $n_{HB}$ is the number of cells connected in series for phase.

Separated DC sources are an advantage in many applications but this feature leads to a more complex DC-voltage regulation loop.

Others topology structures of multilevel inverters and different classification methods were reported in literature [4]. In [5], an interesting classification into two comprehensive categories according to the applied DC source structure were discussed. A high number of topology structures were developed since separated DC sources are very diffused in renewable energy plants (PV, Wind farm, Fuel cell, etc.).

### 1.1. Overview of Pulse Width Modulation (PWM) Modulation Techniques

Pulse width modulation (PWM) techniques found large use in many industrial applications due to their easy implementation in the modern control systems and the high flexibility. Generally, PWM techniques used for multilevel inverters are an extension of modulation techniques for the traditional two-level voltage source inverters (VSI). A general classification of the modulation strategies for MPI presents two categories: "Fundamental switching frequency" and "High switching frequency".

Generally, the first category have been used in application where it is necessary to reduce the switching losses (i.e., high power electrical drive) while the second category have been used in applications where it is necessary to reduce the harmonic content on the output voltage (i.e., grid connected systems).

In literature [6–8] were reported many multicarrier modulation PWM methods, which differ for the reference signals and carrier signals. About the carrier signals, there are the "amplitude shifted" multicarrier PWM strategies and the "phase shifted" multicarrier PWM.

Amplitude shifted multicarrier PWM presents three alternative PWM strategies with the identical peak-to-peak amplitude and different phase relationships between the carriers, which are:

- Phase Disposition (PD) (Figure 2a), where all carriers are in phase;
- Phase Opposition Disposition (POD) (Figure 2b), where the carriers above the reference zero point have a difference of phase respect those below the zero point of $\pi$;
- Alternative Phase Opposition Disposition (APOD) (Figure 2c), where each carrier is phase shifted by $\pi$ from its adjacent carriers.

The carrier number $n_c$ of the level shifted multicarrier PWM in function of the number of the converter level $n_L$, is equal to:

$$n_c = n_L - 1 \tag{2}$$

These strategies lead to elimination of all carriers and related sideband harmonics up to the switching frequency.

Phase shifted multicarrier PWM strategy, shown in Figure 2d, is an extension of the unipolar PWM for traditional single-phase two-level inverter. For this technique, the modulation of the H-bridge inverters in each phase leg is modular. Thus, the reference waveforms for the two-phase legs inverter are phase shifted by $\pi$. The number of the carrier signals is equal to $n_{HB}$ while the phase shifted optimum (PSO) to obtain harmonic cancellation, is achieved:

$$PSO = \frac{(i-1)\pi}{n_{HB}} \tag{3}$$

where $i$ is the $i^{th}$ H-Bridge series connected per phase. For a five-level inverter two carrier signals with mutual phase shift equal to $\pi/2$, Figure 2b, are necessary. This scheme leads to elimination of all carriers and associated sideband harmonics up to the $2n_{HB}$ times of the switching frequency.

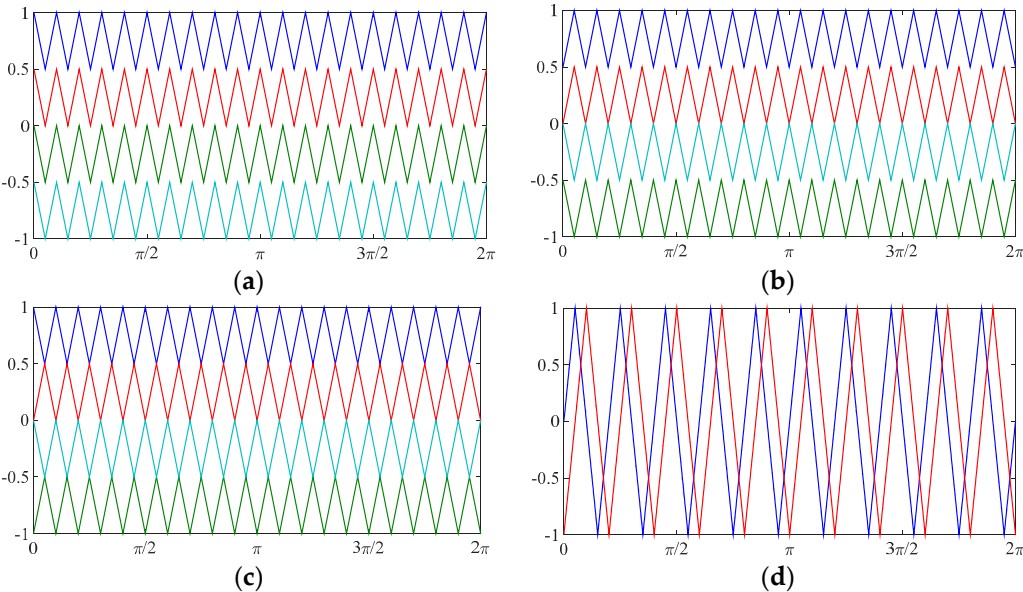

**Figure 2.** Multicarrier strategies for five-level converter: (**a**) Phase Disposition (PD); (**b**) Phase Opposition Disposition (POD); (**c**) Alternative Phase Opposition Disposition (APOD); (**d**) Phase Shifted (PS). For the first three modulation techniques, four carriers are required, for example in PD technique blue and red lines enable the voltage control of higher bridge, green and cyan the lower bridge. In PS each only two carriers are required since each phase leg has a modular control.

About the reference signals, there are three alternative:

1.  Sinusoidal reference;
2.  Third harmonic injection (THI);
3.  Switching frequency optimal (SFO).

The THI allows overcoming the limit of the three-phase inverters about the reduction of the maximum peak fundamental line voltage of $\sqrt{3}V_{DC}/2$ (86.60% of $V_{DC}$). Modulation index can be increased by including a common mode third-harmonic term into the reference signal of each phase leg, as shown in Figure 3a (green curve).

This third-harmonic component does not effect on the fundamental line-to-line voltage because the common mode voltages cancel between the phase legs. According to [9], the optimum third-harmonic injection component must have a magnitude of 25% of the fundamental reference. In this way, it is possible to obtain an increasing of the modulation index up to 1.12 and a maximum value of the peak fundamental line-to-line voltage equal to 97% of $V_{DC}$.

Figure 3b shows the SFO signal (green curve) for a phase of the converter. As demonstrated in [9], SFO is a space vector equivalent reference voltage that can be used in PWM modulation to produce output voltages with the same average low-frequency content.

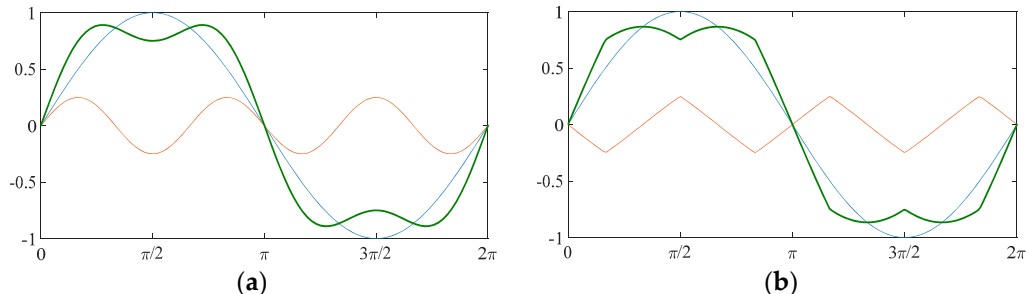

**Figure 3.** Reference signal for a phase of the converter: (**a**) third harmonic injection (THI); (**b**) switching frequency optimal (SFO). Blue waveform represents the fundamental, orange is the adjustment signal and blue is the modified reference.

Like the THI, a sinusoidal reference and the three-times-fundamental-frequency triangular reference, called "voltage offset" $v_{offset}$, compose SFO reference signal. The mathematical expression of the SFO signal for the three-phase system is:

$$\begin{aligned} v_a^*(t) &= v_a(t) - v_{offset} \\ v_b^*(t) &= v_b(t) - v_{offset} \\ v_c^*(t) &= v_c(t) - v_{offset} \end{aligned} \tag{4}$$

where $v_a(t)$, $v_b(t)$ and $v_c(t)$ are the sinusoidal reference that can be expressed in function of the modulation index $M$ as (5):

$$\begin{aligned} v_a(t) &= M\sin(\omega t) \\ v_b(t) &= M\sin\left(\omega t - \frac{2\pi}{3}\right) \\ v_c(t) &= M\sin\left(\omega t - \frac{4\pi}{3}\right) \end{aligned} \tag{5}$$

The voltage offset $v_{offset}$ can be expressed as (6):

$$v_{offset} = \frac{\max(v_a, v_b, v_c) + \min(v_a, v_b, v_c)}{2} \tag{6}$$

The arrangement between carrier signals and modulating references produces twelve modulation techniques, graphically summarized in Figures 4–6.

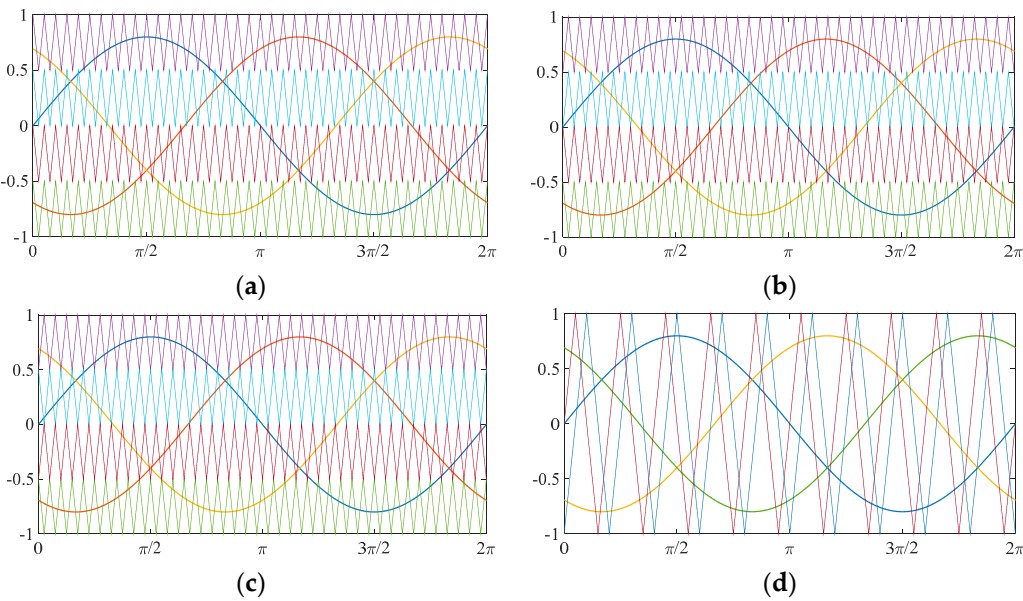

**Figure 4.** Proposed modulation techniques with sinusoidal reference: (**a**) Phase Disposition (PD); (**b**) Phase Opposition Disposition (POD); (**c**) Alternative Phase Opposition Disposition (APOD); and (**d**) Phase Shifted (PS). Blue, red and orange sinusoidal signals represent the reference signals; interferences with the triangular signals generate the modulation angles for the four switches.

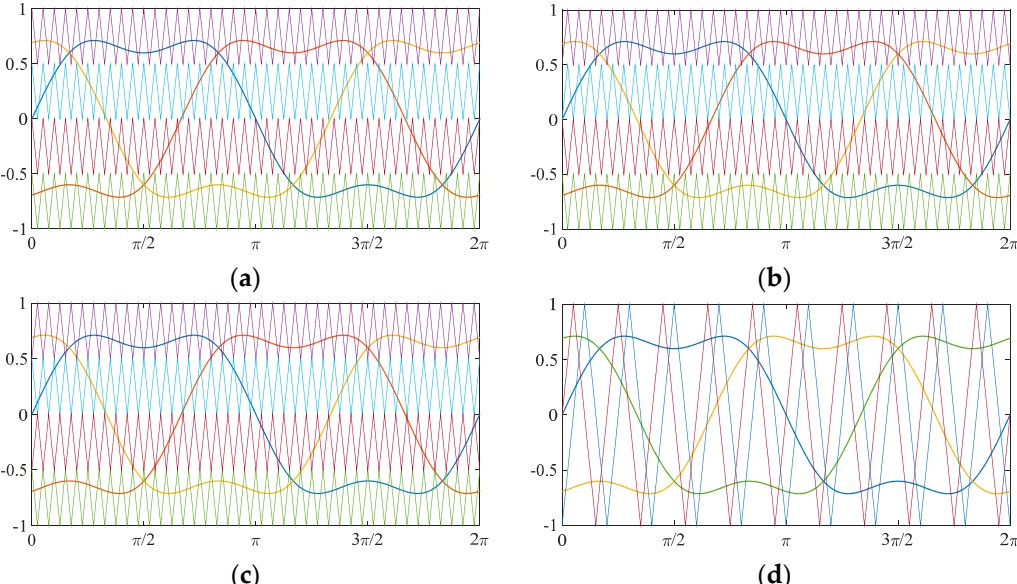

**Figure 5.** Proposed modulation techniques with THI reference: (**a**) Phase Disposition PD; (**b**) Phase Opposition Disposition POD; (**c**) Alternative Phase Opposition Disposition APOD; and (**d**) Phase Shifted PS. Blue, red and orange THI signals represent the reference signals; interferences with the triangular signals generate the modulation angles for the four switches.

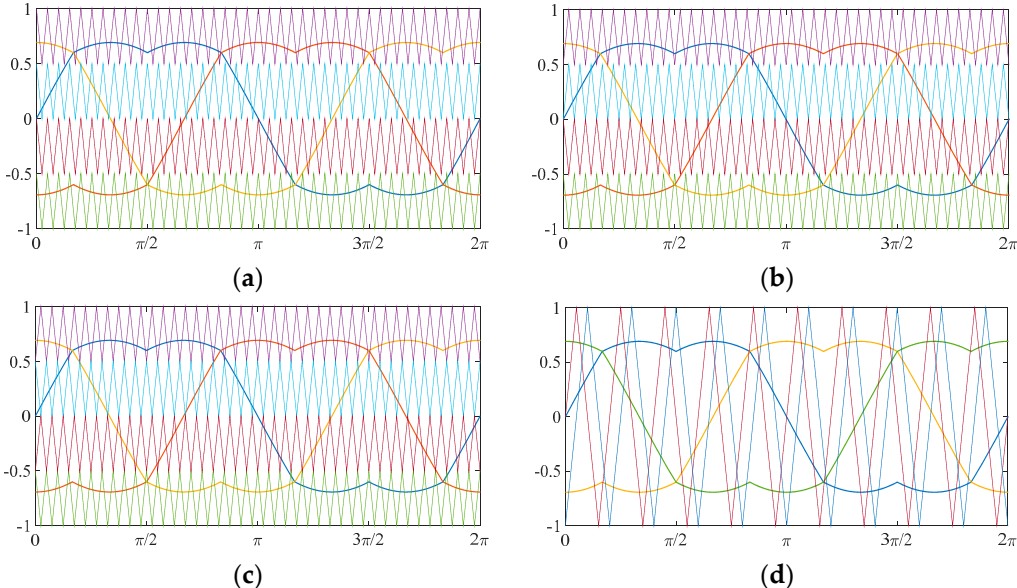

**Figure 6.** Proposed modulation techniques with SFO reference: (**a**) Phase Disposition (PD); (**b**) Phase Opposition Disposition (POD); (**c**) Alternative Phase Opposition Disposition (APOD); and (**d**) Phase Shifted (PS). Blue, red and orange SFO signals represent the reference signals; interferences with the triangular signals generate the modulation angles for the four switches.

An interesting deep discussion on the previous proposed techniques can be found in [10], in which some features of the proposed technique can be found without the control issue and filter design; preliminary simulations of the multicarrier PWM modulation techniques for a three-phase five-level cascaded H-bridge multilevel inverter (CHBMI) were also reported in [11,12]. In these works, the authors addressed that the modulation techniques with sinusoidal reference should present the lower values of the total harmonic distortion rate. A complementary study on the use of B-Spline functions as carrier signals replacing triangular waveforms, can be found in [13,14]. This study confirms that the traditional triangle waveforms as carrier signals are solutions that allow best performances.

### 1.2. Digital Control Boards for Power Converters

The fast technological growth of the electronic design automation (EDA) and the very large scale integration (VLSI) has significantly contributed to the development of programmable digital systems with high performances both in terms of execution time and compactness for the realization of control systems. In addition, the recent advances of software for the implementation, simulation and validation of digital systems, dedicated to the control of specific applications, has contributed to simplify and speed-up the overall design process of the digital controller, which represents the core of modern systems for the electrical energy conversion. Figure 7 shows a block diagram of a typical grid-connected system.

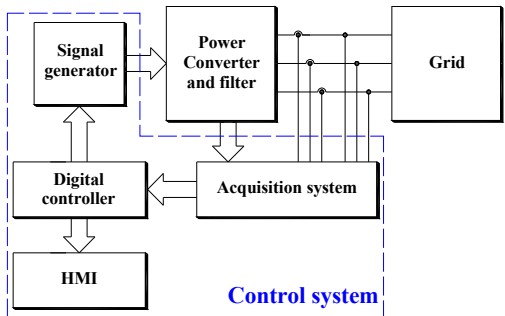

**Figure 7.** Block diagram of a typical grid-connected system. HMI: human–machine interface.

The block named Power Converter and filter requires a control system supervising the behavior (harmonic reduction, filter efficiency, etc.). Generally, the control system consists of four parts:

1.  Acquisition system, which provides signal conditioning and digital acquisition of electrical measures (usually current, voltage, frequency) and other quantities (usually solar irradiation, temperature, etc.);
2.  Digital controller, required for algorithms employment (filtering, identification, control, modulation of output signals and others);
3.  Human–machine interface (HMI), suitable for setup phase as well as for monitoring functions;
4.  Signal generator, allowing conversion of the digital signals to analog signals for the power components.

Digital controller represents the core of the control system, different are the digital controller available in the market. A first example of digital controller is the Microcontrollers or DSP (digital signal processor), allowing the implementation of the control algorithms through a purely software programming (with C or C++). The constructor defines the DSP hardware and it is composed by several peripherals, such as the RAM or the ROM. However, an already designed hardware structure reduces the flexibility in the use of the microcontroller. In fact, a complex issue is related to the time of computing of the control algorithm, due to the fact that all the operation needed for this computing are executed in a sequential routine, causing losses in terms of efficiency of the control system [15].

The FPGA (field programmable gate array) is another example of digital controller, composed by a matrix of configurable logic blocks (CLBs) with completely reprogrammable connections. This fact leads to a higher flexibility with respect to a DSP, allowing the realization of specific hardware structures in dependence of the nature of applications. In addition, this feature allows the realization of a system of logic operations developed in parallel, reducing the time of computing. Thus, by means of an FPGA, high-performance control systems can be realized, even comparable with equivalent controllers composed by analogical components, as reported in [16–18].

For instance, the main advantages provided by the adoption of an FPGA-based system for the current control in AC machine drives is presented in [19]. In particular, the FPGA system significantly reduces the execution time of the control algorithm, increasing, therefore, the performances of the related system.

### 1.3. Literature Survey

Research in the world of multilevel inverter is very wide. Different reviews can be found in the literature [20–23], regarding topologies, switching frequencies, employment of photovoltaic sources (PV), control and cost of inverters, depending on different factors such number of sources, switches, and connections.

Novel structures are continuously suggested [24,25] and ways to control them [26–28]. Efficiency, dimension, weight, and reliability influence the cost of manufacturing inverters. Nowadays, multilevel inverters reached in efficiency the value of 98% and the achievement of the next 1% increase is a hard challenge, and ever-more efficient and advanced modulation techniques are required, which are embracing two different ways: low switching frequency modulation techniques and high switching frequency PWM.

Two systems coexist. The main advantage of the employment of the low switching frequency modulation technique is the reduction of the switching losses to a minimum assessment and the confining the stress on the power components (less overshoot). Through one period of the fundamental reference, low switching frequencies techniques generally act one or two commutations of the switches, so generating a staircase waveform. However, the output voltage waveform has different low order harmonics, with amplitudes similar to the fundamental one, hard to be filtered.

Different are the works that face the issue of reducing the harmonics distortion rate (THD) [29–35]. Some of them exploit the problematic of extraction maximum power from PV modules [29] or

implement innovative fast switching modulation, with or without considering power losses [30,31]. As reference, the authors of [32] used a single-phase multilevel inverter scheme, employing three series connected full-bridge stages and a single half-bridge inverter. The result was a reduction of harmonics, evaluated in terms of total harmonic distortion rate, about 9.85%. Spice models help researchers to consider coupled issues such as THD and storage elements [33]. Again, simulation allows the tracking of the performance of a multilevel inverter in partial shaded condition of PV panels [34]. Simulation seems to be the best way to predict how reduce the THD with grid connected systems [35]. The simulation produces a 15-level output voltage, the total harmonic distortion was about 8.12%, a very low level was reached, but similar multilevel inverters will be overpriced.

On the opposite side, higher frequency modulation presents harmonics with higher frequencies, which requires an economic filters, but high frequency switching brings also higher losses due circulating currents. An evaluation of harmonics content, useful to better define the output voltage waveforms, are reported [10–12], and will be objects of deeper discussions in this paper.

High power electrical drives applications need mainly the reduction of the switching losses and electromagnetic interferences (EMI). Losses concur to define efficiency of converter, so soft switching modulation techniques, employing selective harmonic elimination (SHE) technique and selective harmonic mitigation (SHM) technique, are frequently chosen. SHE method requires the choosing of a single $h^{th}$ harmonic to be removed, so a set of non-linear sinusoidal formulas is solved by choosing the switching angles. The SHM techniques, instead, mitigates simultaneously different harmonics by correctly choosing the switching angles.

Whichever technique is employed, SHE or SHM, to resolve the group of transcendental equations and to discover the related switching angles, different approaches can be taken into account. Obviously, the simplest approach develops iterative methods such as Newton–Rhapson. As reference in [36], the Newton–Rhapson iterative method is employed for the assessment of switching angles for a seven level inverter. The total harmonic distortion of the staircase voltage output is equal to 11.8%.

The authors of [37] present an evaluation between different modulation techniques, applied for a five-level cascaded H-bridge multilevel inverter. The employed control scheme enforces three different pre-defined arrangements for the switching angles. By the uses of these schemes there is an achievement of a minimum THD around 17.07% for the waveform of voltage. The work presented in [38], employs a particularly fast optimal solution of harmonic elimination techniques, used inverter is a five-level multilevel one, and also non-equal DC sources feed the different levels. The solution of the problem is entrusted to a novel particle swarm optimization (PSO) algorithm, and THD achieved a minimum value of 5.44%.

Authors of [39] proposed an optimal SHM technique for a seven-level inverter scheme. The individual harmonic to be mitigated and the THD are subjected to satisfaction of three voltage harmonic standards, named EN50160 [40], CIGRE JWG C4.07 [41] and IEC61000-3-6 [42].

Paper [43] proposes again the PSO technique in presence of PV sources with different voltage levels, the non-linear transcendental equations were solved offline. THD was minimized by employing of pre-calculated switching angles. By the dual employment of the adaptive neuro fuzzy inference system (ANFIS) and also maximum power point tracker (MPPT) algorithm, PV DC sources were transformed in identical DC source. The resulting THD was around 3.7%, less than the ones recommended by IEEE-519 (5%).

Author of [44] again employed an adaptive neuro-fuzzy interference system in order to eliminate voltage harmonics. The proposed comparison shows a best performance of ANFIS referred to neuro fuzzy controller (NFC), in the case of studying a seventh level inverter with active filter. Again the option of an active filter, used in order to improve the performance of the control, can be find in [45].

In [46] the method known as "voltage cancelation" is used for single-phase H-bridge inverters, and in [47] was applied for a single-phase five-level CHBMI.

In this paper the structures used in [47] with a soft switching modulation is used with high switching PWM in order to achieve the same or a better THD.

### 1.4. Contributions and the Organization of Paper

The purpose of this article is to define in detail all the information required to implement the modeling, the operation and the control of a CHBMI.

By starting from general information on CHBMI a detailed report is presented. Informations for the reproduction of the results are not omitted, different tables report the used parameters, from the impedances used in the filtering systems for the various PWM employed techniques (twelve different cases) to the delay times used for driving the system.

Particular attention is devoted to the following aspects: LCL filter design, control design, harmonic content and validation of the proposed approach.

Section 2 is devoted to the simulation of a CHBMI for grid connected applications. The instantaneous model of the converter will be presented in Section 2.1 and after a CHBMI average model. The LCL filter design will be introduced in Section 2.2 and the controller design in Section 2.3. Finally, the simulated performances are evaluated in Section 2.4.

Section 3 is devoted to the experimental validation. The Test Bench will be described in Section 3.1. The control algorithm design will be presented in Section 3.2. The model validation will be discussed in Section 3.3 and finally the grid connected application will be introduced in Section 3.4.

Section 4 recalls and discusses the obtained results, and finally Section 5 concludes the paper.

## 2. Cascaded H-Bridge Inverter for Grid Connected Applications: Modelling and Control

The purpose of this section is to provide all the useful information to realize a virtual model of a CHBMI.

### 2.1. Mathematical Model of the System

Among the classic structures of multilevel inverters presented in literature, this work considers the three-phase, five-level Cascaded H-Bridge inverter topology. Figure 8 shows the topology structure of a three-phase five-level CHB inverter connected to the grid through a LCL filter and a transformer (used for boost voltage and security purpose).

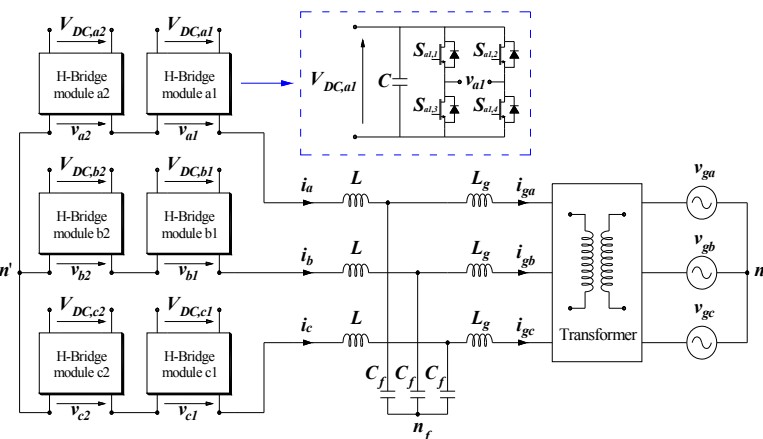

**Figure 8.** Topology structure of a three-phase five-level CHB inverter system.

Each phase of the CHB inverter consists of two cascaded H-bridges in series connected. Thus, the phase voltages $v_a(t)$, $v_b(t)$ and $v_c(t)$, referred on the $n'$ point, is obtained by summing output voltage of the series connected H-Bridges (7):

$$v_a(t) = v_{a1}(t) + v_{a2}(t)$$
$$v_b(t) = v_{b1}(t) + v_{b2}(t) \tag{7}$$
$$v_c(t) = v_{c1}(t) + v_{c2}(t)$$

Taking into account the switching state of the power components, the instantaneous model of the three-phase five-level CHBMI in both the AC and DC side can be totally described by the following equations:

$$
\begin{aligned}
v_a(t) &= V_{DC,a1} \cdot (S_{a1,1} - S_{a1,2}) + V_{DC,a2} \cdot (S_{a2,1} - S_{a2,2}) \\
v_b(t) &= V_{DC,b1} \cdot (S_{b1,1} - S_{b1,2}) + V_{DC,b2} \cdot (S_{b2,1} - S_{b2,2}) \\
v_c(t) &= V_{DC,c1} \cdot (S_{c1,1} - S_{c1,2}) + V_{DC,c2} \cdot (S_{c2,1} - S_{c2,2})
\end{aligned}
\tag{8}
$$

$$
\begin{aligned}
i_{DC,a1} &= i_a \cdot (S_{a1,1} - S_{a1,2}) & i_{DC,a2} &= i_a \cdot (S_{a2,1} - S_{a2,2}) \\
i_{DC,b1} &= i_b \cdot (S_{b1,1} - S_{b1,2}) & i_{DC,b2} &= i_b \cdot (S_{b2,1} - S_{b2,2}) \\
i_{DC,c1} &= i_c \cdot (S_{c1,1} - S_{c1,2}) & i_{DC,c2} &= i_c \cdot (S_{c2,1} - S_{c2,2})
\end{aligned}
\tag{9}
$$

$$
\begin{aligned}
C\frac{dV_{DC,a1}}{dt} &= i_{in,a1} - i_{DC,a1} & C\frac{dV_{DC,a2}}{dt} &= i_{in,a2} - i_{DC,a2} \\
C\frac{dV_{DC,b1}}{dt} &= i_{in,b1} - i_{DC,b1} & C\frac{dV_{DC,b2}}{dt} &= i_{in,b2} - i_{DC,b2} \\
C\frac{dV_{DC,c1}}{dt} &= i_{in,c1} - i_{DC,c1} & C\frac{dV_{DC,c2}}{dt} &= i_{in,c2} - i_{DC,c2}
\end{aligned}
\tag{10}
$$

where $S_{ji,k}$ ($j = a \dots c$; $i,k = 1$ or $2$) are the switching state in which "1" represents that the switch is ON and "0" represents that the switch is OFF.

Equations (8) and (9), can be simplified in (12) and (13) by considering the same DC voltage $V_{DC}$ for each H-Bridges and by defining the switching functions $S_{ji} \in \{-1, 0, 1\}$ as:

$$
S_{ji} = S_{ji,k} - S_{ji,k+1}
\tag{11}
$$

Thus, Equations (14) and (15) can be rewritten as:

$$
\begin{aligned}
v_a(t) &= V_{DC} \cdot (S_{a1} + S_{a2}) \\
v_b(t) &= V_{DC} \cdot (S_{b1} + S_{b2}) \\
v_c(t) &= V_{DC} \cdot (S_{c1} + S_{c2})
\end{aligned}
\tag{12}
$$

$$
\begin{aligned}
i_{DC,a1} &= i_a \cdot S_{a1} & i_{DC,a2} &= i_a \cdot S_{a2} \\
i_{DC,b1} &= i_b \cdot S_{b1} & i_{DC,b2} &= i_b \cdot S_{b2} \\
i_{DC,c1} &= i_c \cdot S_{c1} & i_{DC,c2} &= i_c \cdot S_{c2}
\end{aligned}
\tag{13}
$$

Equations (12) and (13) represent the instantaneous model of the three-phase five-level CHBMI in terms of the phase voltage and DC current. The line-to-line voltage as described by Equation (14):

$$
\begin{aligned}
v_{ab}(t) &= V_{DC} \cdot (S_{a1} + S_{a2} - S_{b1} - S_{b2}) \\
v_{bc}(t) &= V_{DC} \cdot (S_{b1} + S_{b2} - S_{c1} - S_{c2}) \\
v_{ca}(t) &= V_{DC} \cdot (S_{c1} + S_{c2} - S_{a1} - S_{a2})
\end{aligned}
\tag{14}
$$

The model is complete with the equations to describe the LCL filter behavior and the transformer. Thus, the equivalent circuit have to be taken in to account where $r_{TR}$ and $L_{TR}$ ($r_{TR}$ = 25 m$\Omega$ and $L_{TR}$ = 108.23 $\mu$H) represent the short-circuit impedance reported on the low side of the transformer.

Finally, the following equations can be used for the rating the output currents $i_{\{a,b,c\}}$ of the converter (15), grid currents $i_{g\{a,b,c\}}$ (16) and the capacitor voltages $v^*_{\{a,b,c\}}$ (17), where r and $r_g$ are the resistance of the inductance $L$ and $L_g$ of LCL filter.

$$
\begin{aligned}
L\frac{di_a}{dt} &= v_a - i_a r - v_a^* - v_{n_f n'} \\
L\frac{di_b}{dt} &= v_b - i_b r - v_b^* - v_{n_f n'} \\
L\frac{di_c}{dt} &= v_c - i_c r - v_c^* - v_{n_f n'}
\end{aligned}
\tag{15}
$$

$$\left(L_g + L_{TR}\right) \cdot \frac{di_{ga}}{dt} = v_a^* - i_{ga} \cdot \left(r_g + r_{TR}\right) - v_{ga} - v_{nn_f}$$
$$\left(L_g + L_{TR}\right) \cdot \frac{di_{gb}}{dt} = v_b^* - i_{gb} \cdot \left(r_g + r_{TR}\right) - v_{gb} - v_{nn_f} \tag{16}$$
$$\left(L_g + L_{TR}\right) \cdot \frac{di_{gc}}{dt} = v_c^* - i_{gc} \cdot \left(r_g + r_{TR}\right) - v_{gc} - v_{nn_f}$$

$$C_f \frac{dv_a^*}{dt} = i_a - i_{ga}$$
$$C_f \frac{dv_b^*}{dt} = i_b - i_{gb} \tag{17}$$
$$C_f \frac{dv_c^*}{dt} = i_c - i_{gc}$$

where $v_{n_f n'}$ is the voltage between the point $n_f$ and $n'$ and $v_{nn_f}$ is the voltage between the point $n$ and $n_f$ that can be expressed by the Equation (18).

$$v_{n_f n'} = \frac{1}{3}\left[(v_a + v_b + v_c) - r(i_a + i_b + i_c) - L\left(\frac{di_a}{dt} + \frac{di_b}{dt} + \frac{di_c}{dt}\right)\right]$$
$$v_{nn_f} = \frac{1}{3}\left[(v_a^* + v_b^* + v_c^*) - r_g\left(i_{ga} + i_{gb} + i_{gc}\right) - L_g\left(\frac{di_{ga}}{dt} + \frac{di_{gb}}{dt} + \frac{di_{gc}}{dt}\right)\right] \tag{18}$$

In the case of the equilibration system, the voltage $v_{nnf}$ is equal to zero. In order to design the control system, it is necessary to develop an average model of the system. Generally, the average model takes into account the average values in a switching period $T_{sw}$. In this way, the average phase voltages and average currents can be expressed as (19) and (20):

$$\overline{v}_a(t) = \frac{1}{T_{sw}} \int_0^{T_{sw}} V_{DC} \cdot (S_{a1} + S_{a2}) dt$$
$$\overline{v}_b(t) = \frac{1}{T_{sw}} \int_0^{T_{sw}} V_{DC} \cdot (S_{b1} + S_{b2}) dt \tag{19}$$
$$\overline{v}_c(t) = \frac{1}{T_{sw}} \int_0^{T_{sw}} V_{DC} \cdot (S_{c1} + S_{c2}) dt$$

$$\overline{i}_{DC,a1} = \frac{1}{T_{sw}} \int_0^{T_{sw}} i_a \cdot S_{a1} dt \quad \overline{i}_{DC,a2} = \frac{1}{T_{sw}} \int_0^{T_{sw}} i_a \cdot S_{a2} dt$$
$$\overline{i}_{DC,b1} = \frac{1}{T_{sw}} \int_0^{T_{sw}} i_b \cdot S_{b1} dt \quad \overline{i}_{DC,b2} = \frac{1}{T_{sw}} \int_0^{T_{sw}} i_b \cdot S_{b2} dt \tag{20}$$
$$\overline{i}_{DC,c1} = \frac{1}{T_{sw}} \int_0^{T_{sw}} i_c \cdot S_{c1} dt \quad \overline{i}_{DC,c2} = \frac{1}{T_{sw}} \int_0^{T_{sw}} i_c \cdot S_{c2} dt$$

As demonstrated in [46], the equations of the average voltages and average currents in a switching period become (21) and (22):

$$\overline{v}_a(t) = V_{DC} \cdot (m_{a1} + m_{a2})$$
$$\overline{v}_b(t) = V_{DC} \cdot (m_{b1} + m_{b2}) \tag{21}$$
$$\overline{v}_c(t) = V_{DC} \cdot (m_{c1} + m_{c2})$$

$$\overline{i}_{DC,a1} = i_a \cdot m_{a1} \quad \overline{i}_{DC,a2} = i_a \cdot m_{a2}$$
$$\overline{i}_{DC,b1} = i_b \cdot m_{b1} \quad \overline{i}_{DC,b2} = i_b \cdot m_{b2} \tag{22}$$
$$\overline{i}_{DC,c1} = i_c \cdot m_{c1} \quad \overline{i}_{DC,c2} = i_c \cdot m_{c2}$$

where $m_{ji}$ ($j = a \dots c$; $i = 1$ or 2) is the modulation index of each H-Bridge module. Equations (21) and (22) represent the *average model of the converter* in a switching period.

## 2.2. LCL Filter Design

The power quality is an important aspect for grid-connected systems. According to [48,49], there are several limits related to the power injection on the grid that have to be respected, as: voltage unbalance (three-phase inverters should not exceed 3%), DC current injection ($I_{DC} < 0.5\%$ for IEEE 1574 and $I_{DC} < 1\%$ for IEC 61727) and current harmonics. The standard harmonic current limits, defined

by IEEE 1574 and IEC 61727 at the point of common coupling (PCC), are summarized in Table 1. Therefore, the level fixed of the total harmonic distortion (THD%) is <5%.

**Table 1.** Current harmonic limits.

| Harmonic Order, *h* | Limit in % of Rated Current |
|:---:|:---:|
| $h < 11$ | 4.0 |
| $11 \leq h < 17$ | 2.0 |
| $17 \leq h < 23$ | 1.5 |
| $23 \leq h < 35$ | 0.6 |
| $h \geq 35$ | 0.3 |

On the grid side, a LCL-filter to reduce high-order harmonics that can interfere with other equipment is typically adopted [50]. A step-by-step design procedure for an LCL filter has been proposed in [51,52]. The proposed method employs different factors as inputs such the power rating of the converter, the chosen line frequency and obviously the switching frequency.

According to [52,53], the converter side inductance $L$ is defined in order to bound the current ripple produced by the converter. Grid side inductance $L^*_g$ can be determined as a function of $L$, using the index $r$ ($L^*_g = r \cdot L$). While, Capacitor value $C_f$ can be determined as a percentage $x\%$ of the delivered reactive power under rated conditions.

Aim of this section is to design the LCL filter parameters with the step-by-step method for each modulation techniques taken into account. The step-by-step procedure was applied by considering a system with a line voltage of 50 $V_{rms}$, frequency 50 Hz and rated power of 600 W (parameters of the test bench in Laboratory of Electrical Applications-LEAP of the University of Palermo).

In Table 2 are reported the converter side inductance $L$ values for each modulation techniques taken into account. It should be noted that the lower values of $L$ have been obtained with phase disposition based modulation techniques. Moreover, also phase shifted modulation techniques based present interesting results.

**Table 2.** Grid side inductance values.

|  | **PD** | **POD** | **APOD** | **PS** |
|:---:|:---:|:---:|:---:|:---:|
| Sine | 0.260 mH | 0.882 mH | 0.530 mH | 0.371 mH |
| THI | 0.222 mH | 1.938 mH | 0.584 mH | 0.393 mH |
| SFO | 0.260 mH | 0.882 mH | 0.530 mH | 0.371 mH |

Addition, in the design of the grid side inductance $L^*_g$ has been taken into account the inductance of the transformer, thus, can be expressed as $L^*_g = L_g + L_{TR}$. Table 3 reports the current ripple attenuation depending of the $r$ and $x$ values.

**Table 3.** Current Ripple, $r$ and $x$ values.

|  | **Sine** | | | **THI** | | | **SFO** | | |
|:---:|:---:|:---:|:---:|:---:|:---:|:---:|:---:|:---:|:---:|
|  | *r* | *x*% | $i_g/i$ | *r* | *x*% | $i_g/i$ | *r* | *x*% | $i_g/i$ |
| PD | 1.40 | 1.94% | 10.09% | 1.40 | 2.38% | 10.29% | 1.20 | 2.38% | 10.03% |
| POD | 0.40 | 1.94% | 10.21% | 0.10 | 3.25% | 10.85% | 0.40 | 1.94% | 10.21% |
| APOD | 0.90 | 1.50% | 10.81% | 0.60 | 1.94% | 10.51% | 0.90 | 1.50% | 10.18% |
| PS | 1.00 | 1.94% | 10.21% | 1.20 | 1.50% | 10.65% | 1.00 | 1.94% | 10.21% |

Thus, fixing different values of $x$ and $r$, in particular $x\%$ less than 5% according to limit reported in [52], have been evaluated the current ripple for each modulation techniques taken into account. In this way, have been identified the values of $r$ and $x\%$, summarized in Table 3, that allow to obtain a

current ripple approximately then 10%. By using the values reported in Table 3, have been calculated the filter parameters and are summarized in Table 4.

**Table 4.** LCL filters parameters and frequency resonant.

|          | $L$ (mH) | $C_f$ (µF) | $L^*_g$ (mH) | $L_g$ (mH) |
|----------|----------|------------|--------------|------------|
| SPD      | 0.260    | 8.04       | 0.365        | 0.257      |
| SPOD     | 0.882    | 8.04       | 0.352        | 0.244      |
| SAPOD    | 0.530    | 6.21       | 0.477        | 0.369      |
| SPS      | 0.371    | 8.04       | 0.371        | 0.263      |
| THIPD    | 0.222    | 9.86       | 0.311        | 0.203      |
| THIPOD   | 1.938    | 13.47      | 0.193        | 0.085      |
| THIAPOD  | 0.584    | 8.04       | 0.350        | 0.242      |
| THIPS    | 0.393    | 6.21       | 0.471        | 0.363      |
| SFOPD    | 0.260    | 9.86       | 0.313        | 0.204      |
| SFOPOD   | 0.882    | 8.04       | 0.352        | 0.244      |
| SFOAPOD  | 0.530    | 6.21       | 0.424        | 0.316      |
| SFOPS    | 0.371    | 8.04       | 0.371        | 0.263      |

An interesting consideration is about the PD based modulation techniques because the converter side inductance $L$ and grid side inductance $L_g$ present the lower values. Higher value of the converter side inductance were obtained with THIPOD modulation technique. This phenomenon is attributable at the higher number of the side band harmonics generated by POD carrier signals. In fact, also SPOD and SFOPOD present higher values of the converter side inductance compared others modulation techniques. Regarding the capacitor filter values $C_f$, it is interesting to note that have been obtained similar values among the modulation techniques taken into account.

As stated earlier, it is necessary to evaluate the limits on the parameter values, introduced in step-by-step method, to verify the filter effectiveness. In Table 5 are reported the limits on the LCL parameters values for each modulation techniques.

**Table 5.** Limits on the LCL parameter values.

|          | $\Sigma L$ (p.u.) | $x$ (%) | $f_{res}$ (kHz) |
|----------|-------------------|---------|-----------------|
| SPD      | 0.025             | 1.94%   | 4.54            |
| SPOD     | 0.050             | 1.94%   | 3.53            |
| SAPOD    | 0.041             | 1.50%   | 4.02            |
| SPS      | 0.030             | 1.94%   | 4.11            |
| THIPD    | 0.021             | 2.38%   | 4.44            |
| THIPOD   | 0.087             | 3.25%   | 3.26            |
| THIAPOD  | 0.038             | 1.94%   | 3.79            |
| THIPS    | 0.035             | 1.50%   | 4.35            |
| SFOPD    | 0.023             | 2.38%   | 4.24            |
| SFOPOD   | 0.050             | 1.94%   | 3.53            |
| SFOAPOD  | 0.039             | 1.50%   | 4.15            |
| SFOPS    | 0.030             | 1.94%   | 4.11            |

Each LCL configuration designed allows to respect the limits introduced in step-by-step method. In the next section, is reported the design of the control system for each modulation techniques taken into account.

## 2.3. Controller Design

The objective of the control strategy is to guarantee the synchronization with the main grid and regulate the power injection through the current control loop. Moreover, the current loop is accountable of the power quality issues and of protection for high values of current. Thus, low harmonic content on the current and dynamic response are the important properties of the control system.

According to [54–56], the control strategy is based on the synchronous reference frame, also called *dq* control. Figure 9 shows the schematic block diagram of the *dq* control.

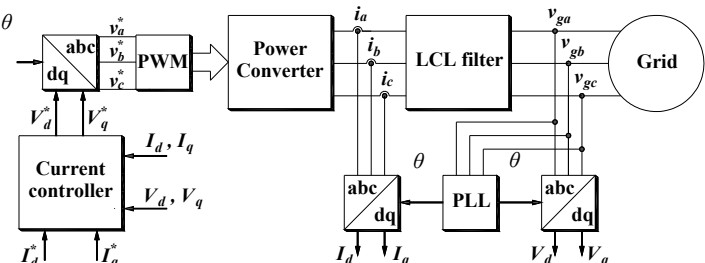

**Figure 9.** Synchronous reference frame control strategy.

The system needs the measurement of the current and voltage through the sensors. Figure 9 shows that the current sensors are on the converter side, since in more applications they are also employed to protect the power converter. Moreover, the LCL filter design and the control design are influenced by the position of the sensors [51].

Synchronous reference frame control is based on the Park's transformation to express both grid currents and voltages into a reference structure rotating synchronously with the grid frequency. In this way, the *dq* components of the voltage and current assume continuous trend and it is possible to use the PI regulator.

The phase-locked loop (PLL) technique [55], allows extracting the instantaneous phase angle of the grid voltage in order to synchronize the voltage waveforms.

For the control design, the instantaneous model of the system, Equations (15) and (16), can be simplified neglecting the filter capacitor $C_f$.

Using the Park's transformation, the Equations (15) and (16) can be rewritten as (23):

$$
\begin{aligned}
v_d(t) &= r_T i_d(t) + L_T \frac{di_d(t)}{dt} - \omega_g L_T i_q(t) + v_{gd}(t) \\
v_q(t) &= r_T i_q(t) + L_T \frac{di_q(t)}{dt} + \omega_g L_T i_d(t) + v_{gq}(t)
\end{aligned}
\tag{23}
$$

where $r_T$ is the sum of the internal resistance of the inductors ($r_T = r + r_g + r_{TR}$) and the $L_T$ is the sum of the inductance of the LCL filter and the short-circuit inductance reported on the low side of the transformer ($L_T = L + L_g + L_{TR}$).

In synchronous reference frame, it is possible to control the *dq* components independently thanks the decoupling of the two-channel control. In this way, through the *d* component it is possible to control the active power while through the *q* component it is possible to control the reactive power.

For the design of the proportional-integral (PI) regulators, the method used to tune the parameters of the PI is the "technical optimum" criterion where both plant and PI regulator have the same time constant in order to simplify the closed loop transfer function.

By according the PI integrator time constant $T_I$ equal to the plant time constant $T_I = L_T/r_T$, with the aim to delete the slower plant pole and supposing a perfect pole-zero cancellation, the current closed-loop transfer function $W(s)$ become of the second order Equation (24).

$$
W(s) = \frac{\frac{k_p}{1.5 L_T T}}{s^2 + \frac{1}{1.5T} s + \frac{k_p}{1.5 L_T T}}
\tag{24}
$$

where $k_p$ is the proportional gain, $T_I$ is the integral time constant and $T$ is the sampling period.

By the analysis of Equation (24), the proportional gain $k_p$ depends on the inductance of the LCL filter. In this way, it is possible to evaluate the PI regulators parameters for each modulation techniques choosing a damping coefficient equal to 0.707. In Table 6 are reported the PI regulator parameters for each modulation techniques taken into account.

**Table 6.** PI regulator parameters.

| | Sine | | | THI | | | SFO | | |
|---|---|---|---|---|---|---|---|---|---|
| | $k_p$ | $T_I$ (ms) | $k_i$ | $k_p$ | $T_I$ (ms) | $k_i$ | $k_p$ | $T_I$ (ms) | $k_i$ |
| PD | 2.09 | 15.65 | 133.73 | 1.78 | 13.36 | 133.73 | 1.91 | 14.35 | 133.73 |
| POD | 4.12 | 30.88 | 133.73 | 7.11 | 53.31 | 133.73 | 4.11 | 30.88 | 133.73 |
| APOD | 3.36 | 25.20 | 133.73 | 3.12 | 23.36 | 133.73 | 3.18 | 23.87 | 133.73 |
| PS | 2.48 | 18.58 | 133.73 | 2.88 | 21.63 | 133.73 | 2.48 | 18.58 | 133.73 |

In the next section, simulation results were reported to evaluate the performance of the system with each modulation techniques taken into account.

### 2.4. Performances Evaluation

Aim of this section is to evaluate the performance of the system through a simulation analysis for each modulation techniques taken into account. In particular, the main purpose is to investigate the effectiveness of the control strategy and the LCL filter in terms of the harmonic content in the currents and voltages in order to determinate the best solution for grid connected applications. The simulation have been carried out in Matlab/Simulink® (version 4.1.1, The MathWorks, Inc., Natick, MA, USA) environment with the same parameters used for each modulation techniques taken into account and reported in Table 7.

**Table 7.** Simulation parameters.

| Electric parameter | Value |
|---|---|
| Grid line Voltage | 50 V |
| Rated current | 6 A |
| Grid frequency | 50 Hz |
| DC Voltages | 24 V |
| Switching frequency | 10 kHz |
| Inductance and resistance of the transformer (low side reported) | 108.23 µH 25 mΩ |

LCL filter requirements and parameters of regulators were determined in the precedent sub-sections (2.2. LCL Filter Design and 2.3. Controller Design). In the follow are reported the results obtained in simulation analysis for each modulation techniques taken into account and have been compared the results among the modulation with the same carrier signals.

2.4.1. Phase Disposition

Generally, the main characteristics of the modulation techniques with PD as carrier signals is that the harmonic spectra of the phase voltage presents a predominant harmonic centered on the switching frequency and side band harmonics. Thus, the difference among SPD, THIPD, and SFOPD are in the amplitude on the harmonic centered on the switching frequency and side band harmonics as shown in Figure 10. However, THI and SFO introduce a third harmonic component on the phase voltage that disappear on the line voltage (three-wired systems).

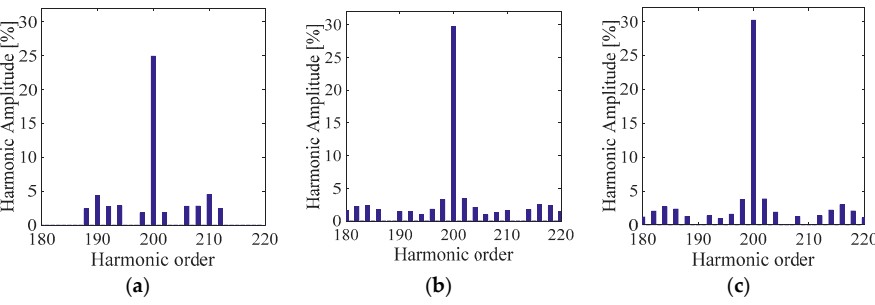

**Figure 10.** Phase voltage harmonic spectra centered around the switching frequency (10kHz) in percent respect to the fundamental amplitude of (**a**) SPD, (**b**) THIPD and (**c**) SFOPD.

This phenomenon explains the different values of THD% and different LCL filter requirements. Consequently, the performance in terms of the harmonic content on the current will be different. Figures 11–13 show the transitory of the converter side currents and grid side currents from zero to the rated current obtained with SPD, THIPD, and SFOPD, respectively. As mentioned above, are visible little differences in the currents trend among SPD, THIPD, and SFOPD, as emphasized by red and green zoom windows. In particular, these differences are present in terms of harmonic content around the switching frequency and the low order harmonics.

Overall, the three-grid side current have a total harmonic distortion—THD% less then 5% according to IEEE 1574 and IEC 61727.

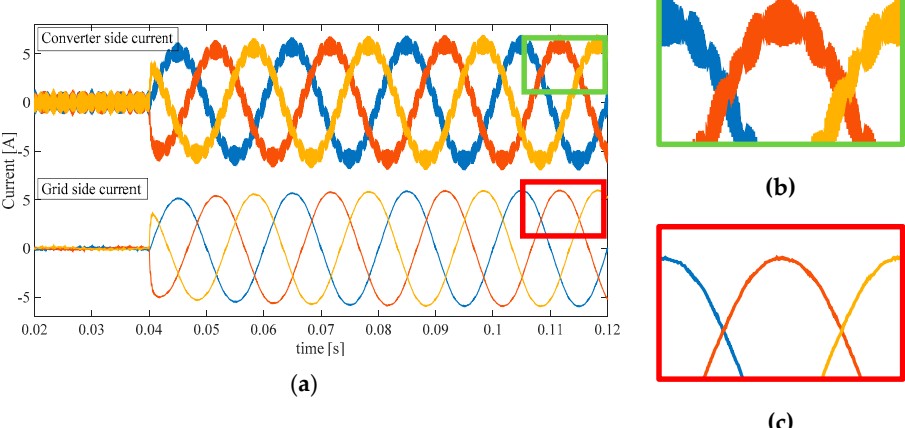

**Figure 11.** Converter side and grid side three-phase current with SPD. (**a**) Transient behavior in multiple cycles; (**b**) Ripple magnification for converter side current; (**c**) Ripple magnification for grid side current.

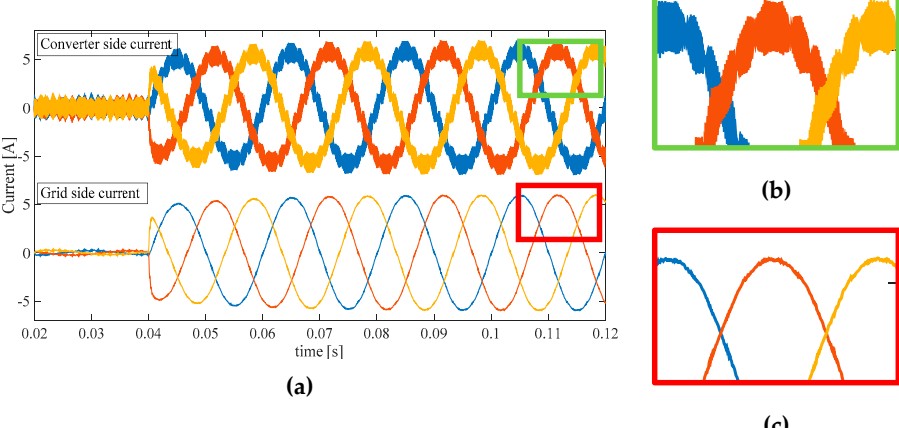

**Figure 12.** Converter side and grid side three-phase current with THIPD. (**a**) Transient behavior in multiple cycles; (**b**) Ripple magnification for converter side current; (**c**) Ripple magnification for grid side current.

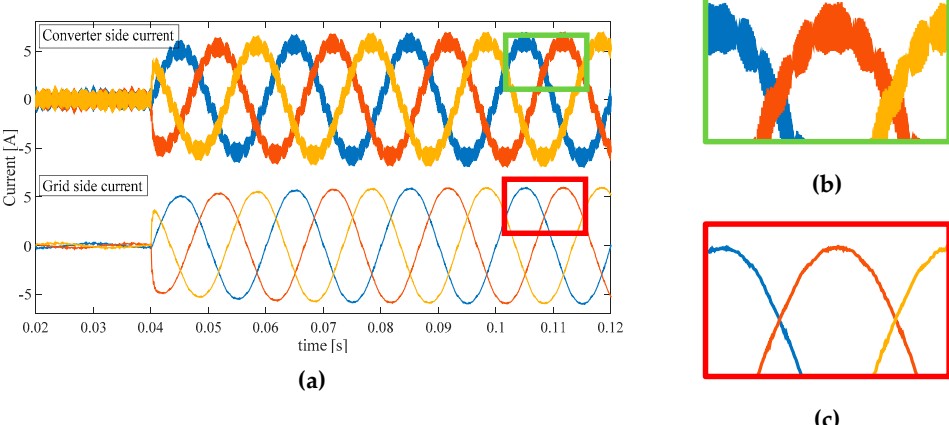

**Figure 13.** Converter side and grid side three-phase current with SFOPD. (**a**) Transient behavior in multiple cycles; (**b**) Ripple magnification for converter side current; (**c**) Ripple magnification for grid side current.

Figure 14 shows the low order harmonics (from third to fortieth harmonic) in grid side current $I_{ga}$ obtained with SPD, THIPD, and SFOPD, respectively.

In the first all, it is interesting to note that the amplitude of the lower order harmonics are below of the standard harmonic current limits defined by IEEE 1574 and IEC 61727 at the PCC. However, is visible only a predominant fifth harmonic in low order spectra obtained with SPD while there are also the eleventh and thirteenth harmonics in the low order spectra obtained with THIPD and SFOPD.

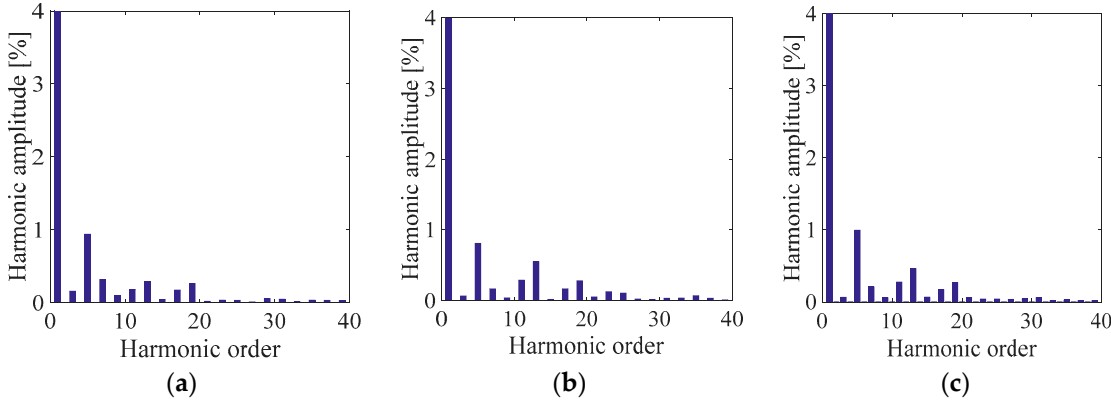

**Figure 14.** Low order harmonics on the grid side current $I_{ga}$ for (**a**) SPD, (**b**) THIPD, and (**c**) SFOPD.

Figure 15 shows a comparison among harmonic spectra centered around the switching frequency of 10 kHz among line voltage $V_{ab}$ (blue bars), converter side current $I_a$ (red bars) and grid side current $I_{ga}$ (yellow bars) obtained with SPD, THIPD, and SFOPD, respectively. The lower values of the grid side current harmonics, less of 0.3% referred to the fundamental amplitude, demonstrate the LCL filter effectiveness. It should be noted that in the spectra of the line voltage and currents appear only side band harmonics thanks to the three-wired connection.

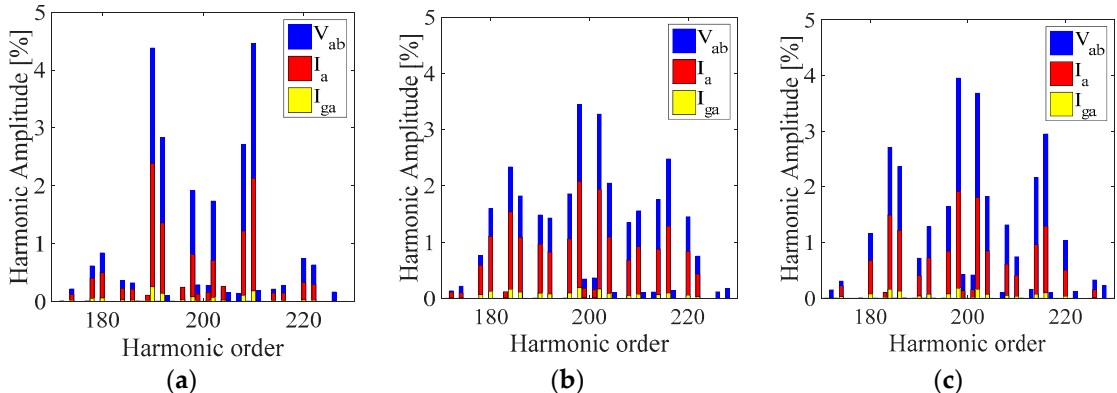

**Figure 15.** Comparison of line voltage harmonic spectra $V_{ab}$, converter side current $I_a$ and grid side current $I_{ga}$ centered around the switching frequency (10 kHz) in percent respect to the fundamental amplitude for (**a**) SPD, (**b**) THIPD, and (**c**) SFOPD.

In particular, harmonic spectra obtained with SPD modulation technique presents three-pair predominant of the side band harmonics while the harmonic spectra obtained with THIPD and SFOPD are different. The tool used to compare the harmonic content around the switching frequency among the SPD, THIPD, and SFOPD is the "Partial Total Harmonic Distortion" (*PTHD*%) defined as:

$$PTHD\% = \frac{\sqrt{\sum\limits_{f_{SW}-n/2}^{f_{SW}+n/2} V_h^2}}{V_1} \cdot 100 \tag{25}$$

where $f_{SW}$ is the switching frequency, $n$ is the bandwidth centered around the switching frequency, $h$ is the harmonic order and $V_1$ is the fundamental amplitude.

In Table 8 are summarized the *PTHD*% calculated for line voltage $V_{ab}$, converter side current $I_a$ and grid side current $I_{ga}$.

**Table 8.** Partial Total Harmonic Distortion" (*PTHD*%) values obtained with SPD, THIPD, and SFOPD.

|  | $V_{ab}$ | $I_a$ | $I_{ga}$ |
|---|---|---|---|
| SPD | 7.94% | 3.92% | 0.38% |
| THIPD | 7.86% | 4.61% | 0.43% |
| SFOPD | 8.29% | 4.08% | 0.38% |

It is interesting to note that the SPD and SFOPD present the lower values of the *PTHD*% as regard to the currents. While, the lower values of the *PTHD*% of the line voltage has been obtained with THIPD modulation technique. Obviously, both the modulation technique and the LCL filter concur to define the harmonic content.

2.4.2. Phase Opposition Disposition and Alternative Phase Opposition Disposition

In harmonic spectra obtained with modulation techniques based POD or APOD as carrier signals, the harmonic component at switching frequency does not appear but there are only side band.

Figure 16 shows the harmonic spectra of the phase voltage centered around the switching frequency obtained with SPOD, THIPOD, SFOPOD, SAPOD, THIAPOD, and SFOAPOD, respectively.

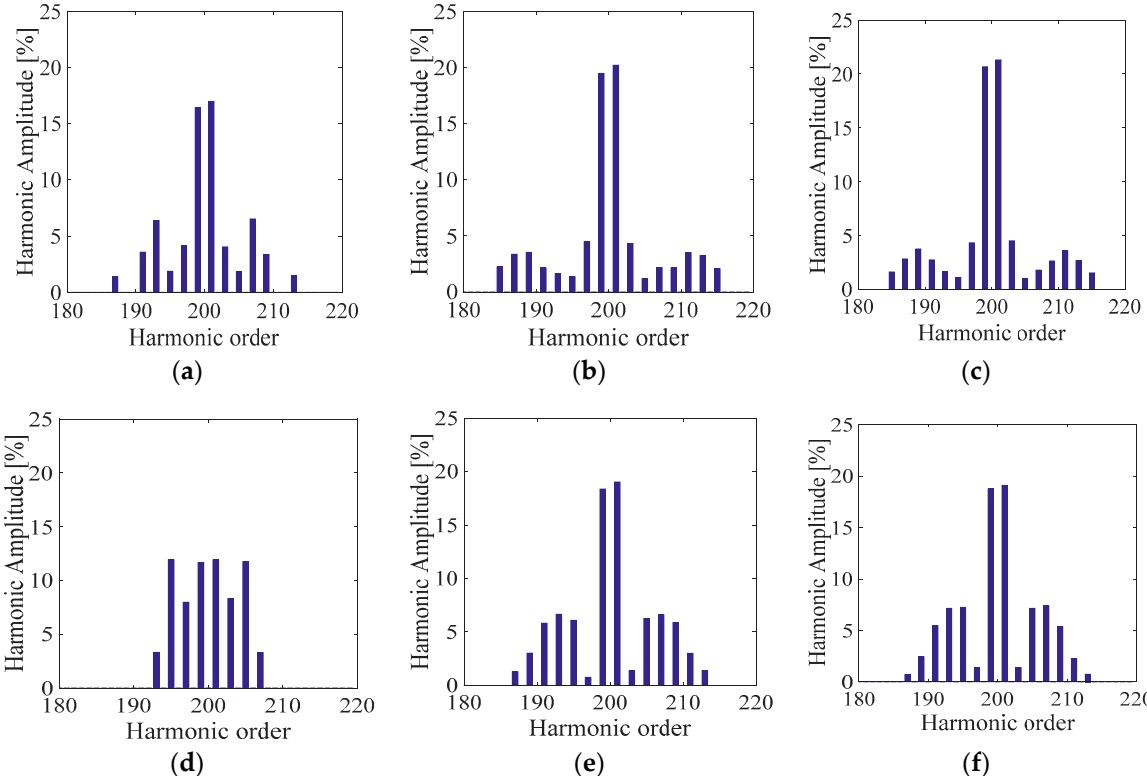

**Figure 16.** Phase voltage harmonic spectra centered around the switching frequency (10kHz) in percent respect to the fundamental amplitude of (**a**) SPOD, (**b**) THIPOD, (**c**) SFOPOD, (**d**) SAPOD, (**e**) THIAPOD, and (**f**) SFOAPOD.

As shown in Figure 16, the harmonic spectra are similar with a pair component predominant respect to the others while only the SAPOD (Figure 16d) presents little differences. Respect the modulation techniques PD or PS based, in the sub-section 2.4. LCL Filter Design the higher values of the filter requirements were obtained with POD and APOD.

Figures 17–22 show the transitory of the converter side currents and grid side currents from zero to the rated current obtained with SPOD, THIPOD, SFOPOD, SAPOD, THIAPOD, and SFOAPOD respectively.

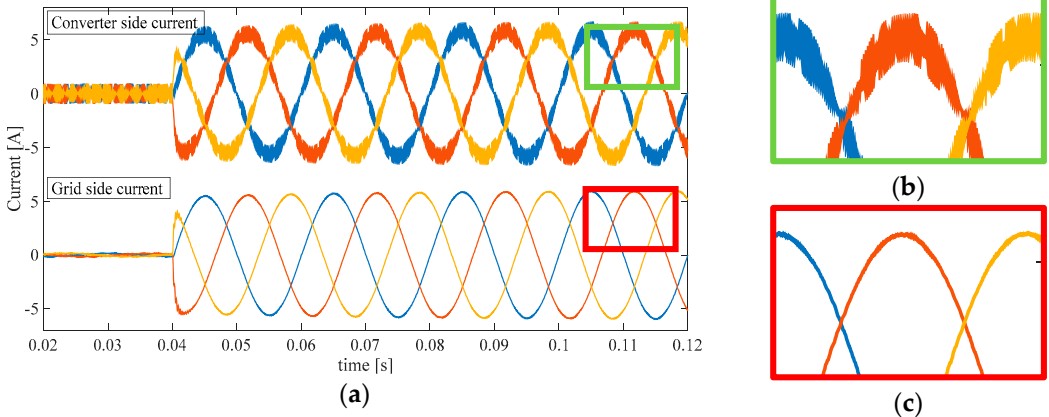

**Figure 17.** Converter side and grid side three-phase current with SPOD. (**a**) Transient behavior in multiple cycles; (**b**) Ripple magnification for converter side current; (**c**) Ripple magnification for grid side current.

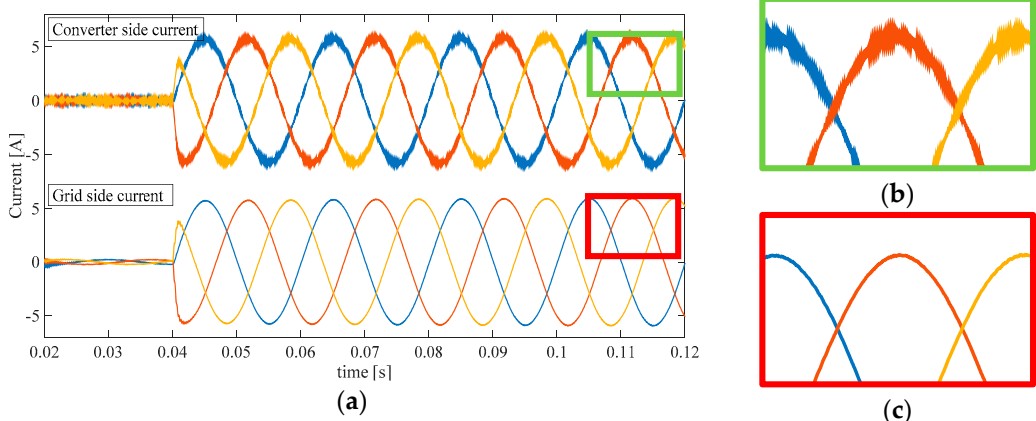

**Figure 18.** Converter side and grid side three-phase current with THIPOD. (**a**) Transient behavior in multiple cycles; (**b**) Ripple magnification for converter side current; (**c**) Ripple magnification for grid side current.

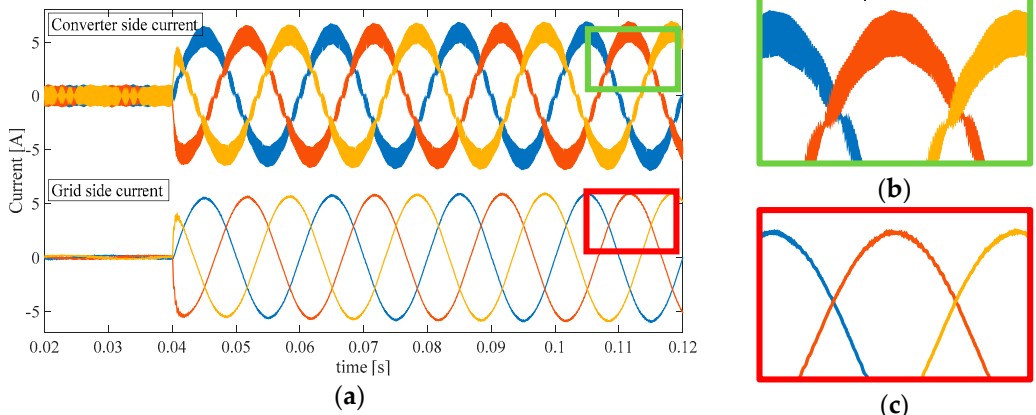

**Figure 19.** Converter side and grid side three-phase current with SFOPOD. (**a**) Transient behavior in multiple cycles; (**b**) Ripple magnification for converter side current; (**c**) Ripple magnification for grid side current.

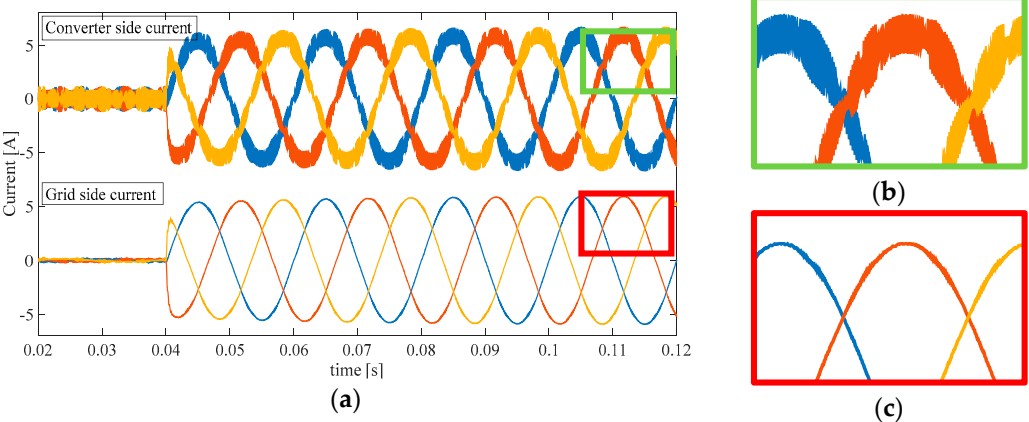

**Figure 20.** Converter side and grid side three-phase current with SAPOD. (**a**) Transient behavior in multiple cycles; (**b**) Ripple magnification for converter side current; (**c**) Ripple magnification for grid side current.

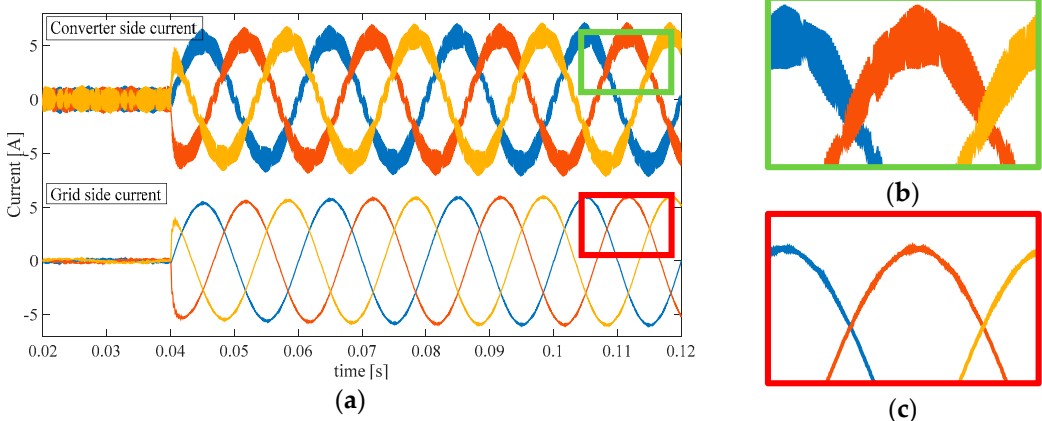

**Figure 21.** Converter side and grid side three-phase current with THIAPOD. (**a**) Transient behavior in multiple cycles; (**b**) Ripple magnification for converter side current; (**c**) Ripple magnification for grid side current.

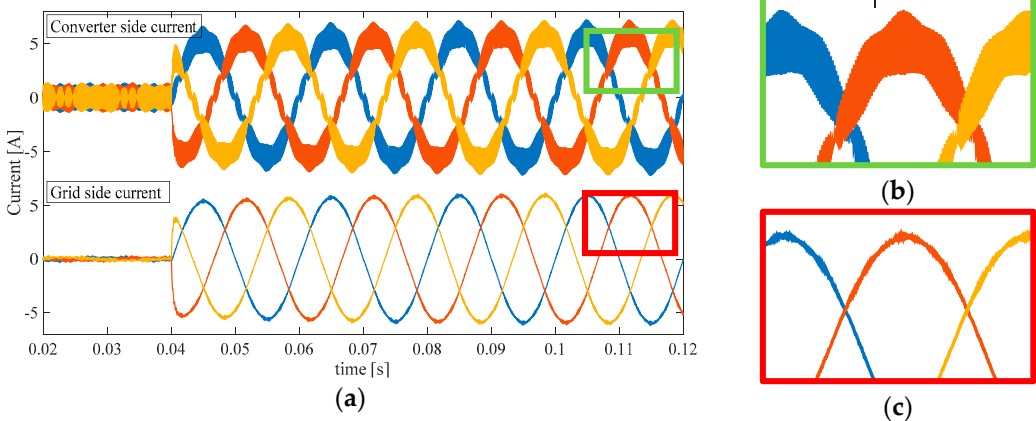

**Figure 22.** Converter side and grid side three-phase current with SFOAPOD. (**a**) Transient behavior in multiple cycles; (**b**) Ripple magnification for converter side current; (**c**) Ripple magnification for grid side current.

As before, the grid currents trend presents a THD% less then 5% according to IEEE 1574 and IEC 61727, but it is necessary to study the lower order harmonics and the ones around the switching frequency, in order to investigate in depth the performances. Figure 23 shows the low order harmonics (from third to fortieth harmonic) in grid side current $I_{ga}$ obtained with SPOD, THIPOD, SFOPOD, SAPOD, THIAPOD, and SFOAPOD, respectively.

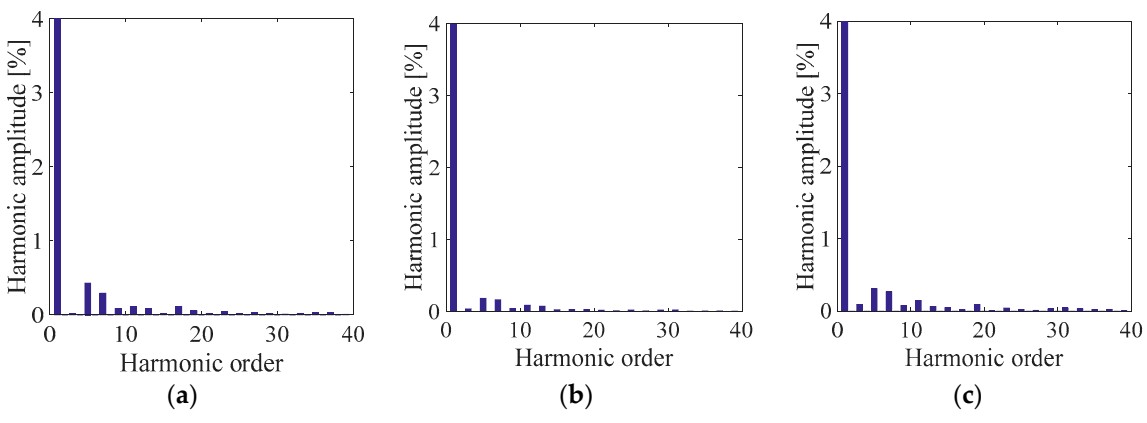

**Figure 23.** *Cont*.

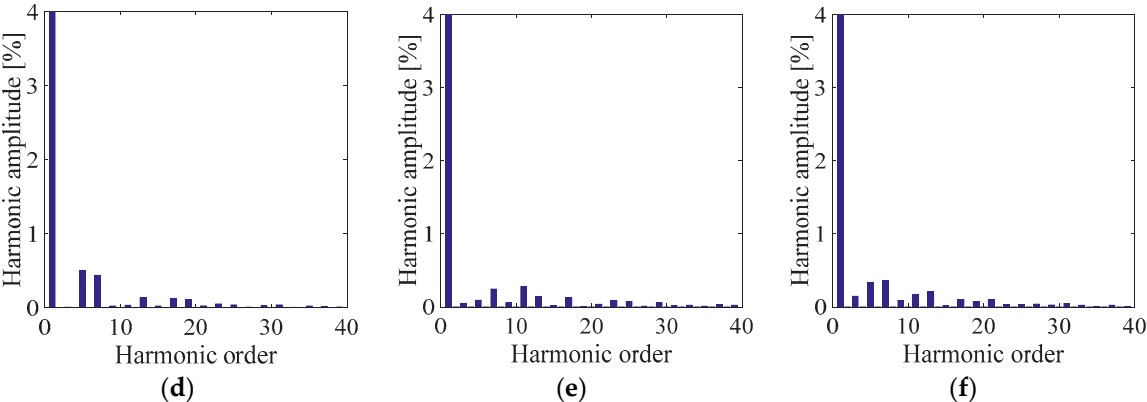

**Figure 23.** Low order harmonics on the grid side current *Iga* for (**a**) SPOD, (**b**) THIPOD, (**c**) SFOPD, (**d**) SAPOD, (**e**) THIAPOD, and (**f**) SFOAPOD.

It should be noted that the amplitude of the lower order harmonics are below of the standard harmonic current limits defined by IEEE 1574 and IEC 61727 at the PCC. Interesting results were obtained with THIPOD (Figure 23b) considering that are visible only fifth and seventh components with the lower amplitude compared to others modulation techniques POD and APOD based. Moreover, this is the best results also compared with modulation techniques PD based. This phenomenon is also explained by the higher values of the LCL filter parameters.

Figure 24 shows a comparison among harmonic spectra centered around the switching frequency of 10 kHz among line voltage $V_{ab}$ (blue bars), converter side current $I_a$ (red bars) and grid side current $I_{ga}$ (yellow bars) obtained with SPOD, THIPOD, SFOPOD, SAPOD, THIAPOD, and SFOAPOD, respectively.

The lower values of the grid side current harmonics, less of 0.3% referred to the fundamental amplitude, demonstrate the LCL filter effectiveness. While, the predominant pair of the side band harmonic that appear in all harmonics spectra explains the higher values of the LCL filter parameters compared with modulation techniques PD or PS based. An interesting consideration is about the grid side current, the values are very lower respect to the modulation techniques PD based.

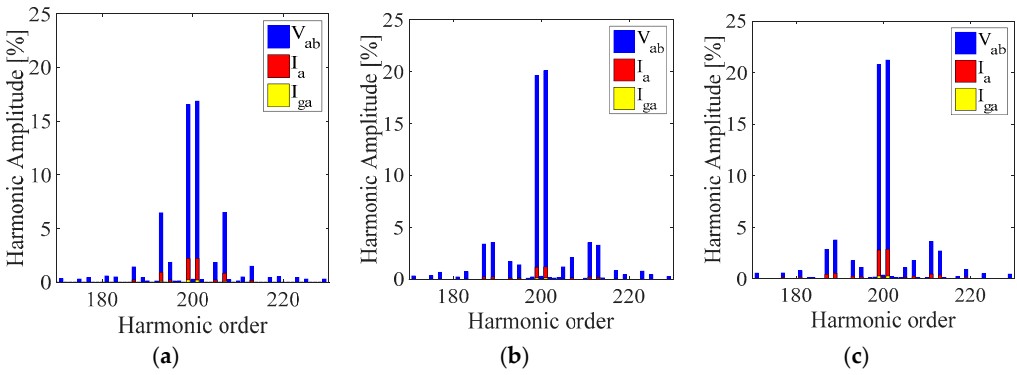

**Figure 24.** *Cont.*

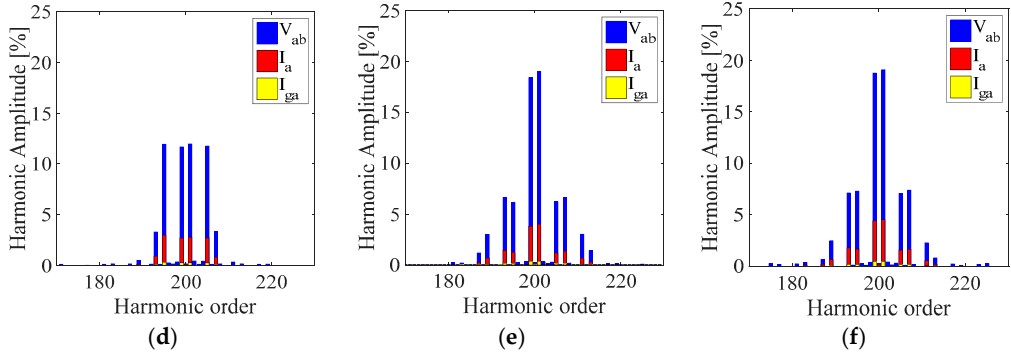

**Figure 24.** Comparison of line voltage harmonic spectra $V_{ab}$, converter side current $I_a$ and grid side current $I_{ga}$ centered around the switching frequency (10 kHz) in percent respect to the fundamental amplitude for (**a**) SPOD, (**b**) THIPOD, (**c**) SFOPOD, (**d**) SAPOD, (**e**) THIAPOD, and (**f**) SFOAPOD.

In Table 9 are reported the values of the PTHD% for line voltage $V_{ab}$, converter side current $I_a$ and grid side current $I_{ga}$.

**Table 9.** *PTHD*% values obtained with SPOD, THIPOD, SFOPD, SAPOD, THIAPOD, and SFOAPOD.

|  | $V_{ab}$ | $I_a$ | $I_{ga}$ |
|---|---|---|---|
| SPOD | 25.62% | 3.37% | 0.33% |
| THIPOD | 29.12% | 1.72% | 0.18% |
| SFOPOD | 30.60% | 4.12% | 0.40% |
| SAPOD | 24.17% | 5.68% | 0.53% |
| THIAPOD | 29.78% | 6.16% | 0.61% |
| SFOAPOD | 30.65% | 7.21% | 0.77% |

As expected, the *PTHD*% relatively of the line voltages present higher values compared with modulation techniques PD based. While, the low values of *PTHD*% relatively of the currents are similar respect to the values calculated with modulation techniques PD based.

2.4.3. Phase Shifted Disposition

Last type of the carrier signals analyzed in this work is the Phase Shifted PS that allows shifting the harmonic to 2*nHB* times of the switching frequency. In the case of the five-level converter, *nHB* is equal to 2, so the harmonic content is shifting to four times the switching frequency (40 kHz). Moreover, the harmonics are centered around four times the switching and are present only side band harmonics like in modulation techniques POD and APOD based. Figure 25 shows the harmonic spectra, centered around four times of the switching frequency, obtained with SPS, THIPS, and SFOPS, respectively.

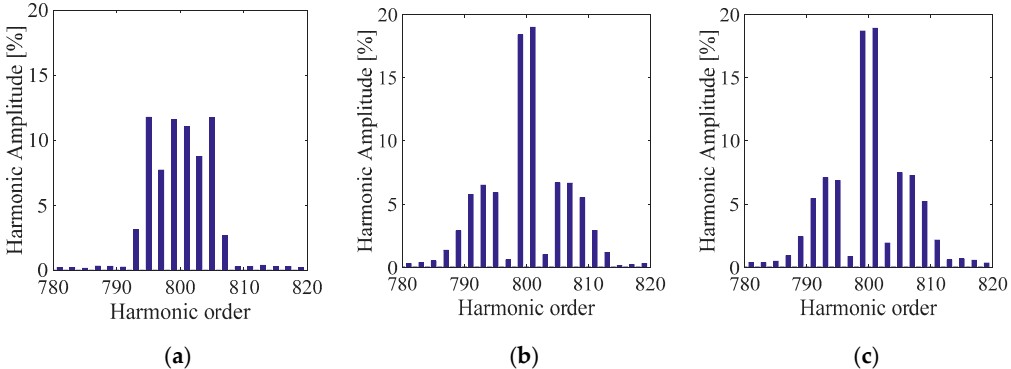

**Figure 25.** Phase voltage harmonic spectra centered around four times of the switching frequency (10 kHz) in percent respect to the fundamental amplitude of (**a**) SPS, (**b**) THIPS, and (**c**) SFOPS.

The harmonic spectra are similar respect to the harmonic spectra obtained with POD and APOD but the amplitude of the predominant harmonics are lower.

Figures 26–28 show the transitory of the converter side currents and grid side currents from zero to the rated current obtained with SPS, THIPS, and SFOPS, respectively.

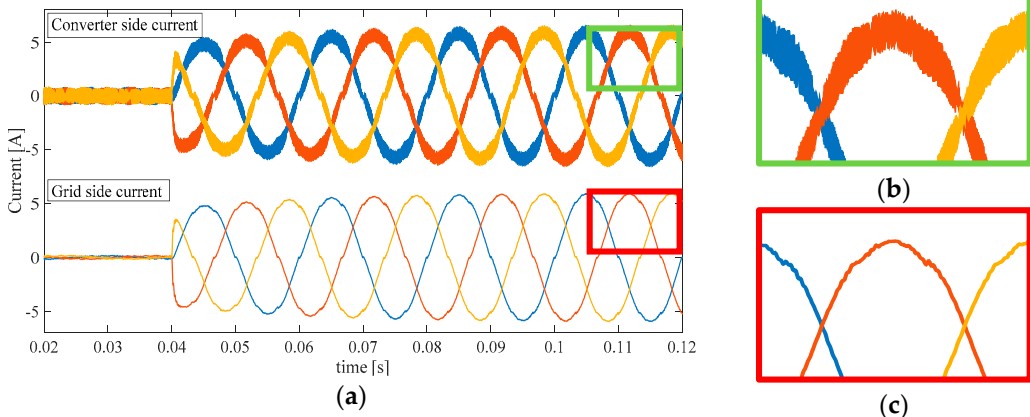

**Figure 26.** Converter side and grid side three-phase current with SPS. (**a**) Transient behavior in multiple cycles; (**b**) Ripple magnification for converter side current; (**c**) Ripple magnification for grid side current.

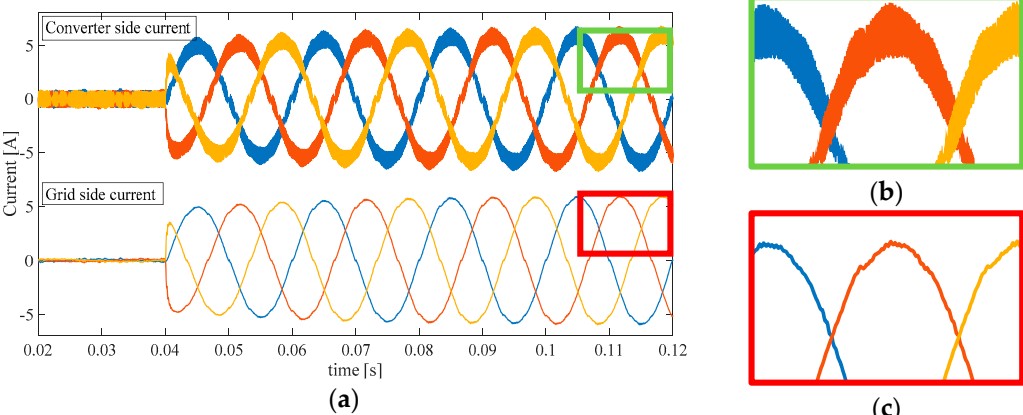

**Figure 27.** Converter side and grid side three-phase current with THIPS. (**a**) Transient behavior in multiple cycles; (**b**) Ripple magnification for converter side current; (**c**) Ripple magnification for grid side current.

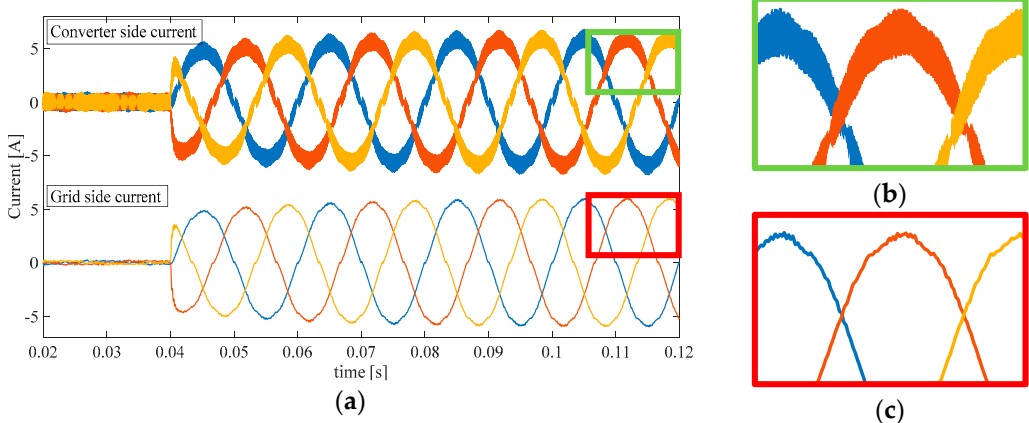

**Figure 28.** Converter side and grid side three-phase current with SFOPS. (**a**) Transient behavior in multiple cycles; (**b**) Ripple magnification for converter side current; (**c**) Ripple magnification for grid side current.

As previously noted, the grid currents trend presents a THD% less then 5% according to IEEE 1574 and IEC 61727 but observing the grid currents trend is evident the presence of low order harmonics.

Figure 29 shows the low order harmonics (from third to fortieth harmonic) in grid side current $I_{ga}$ obtained with SPS, THIPS, and SFOPS, respectively.

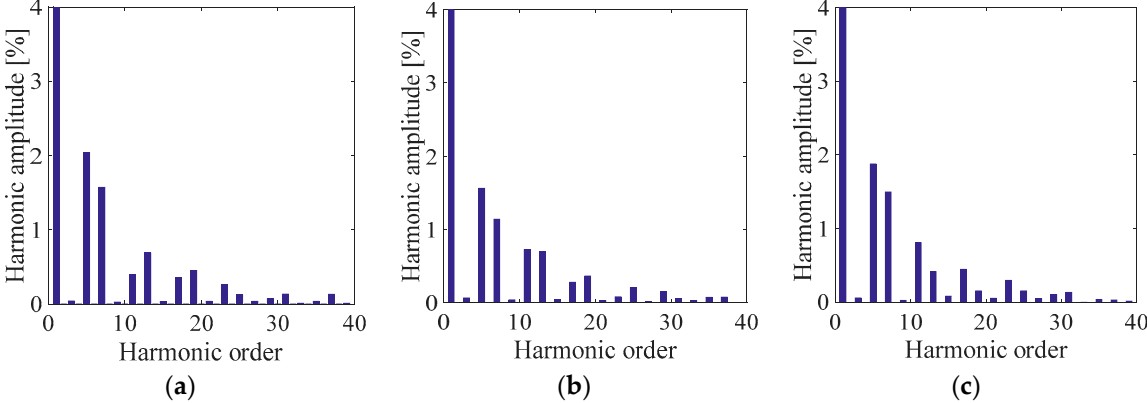

**Figure 29.** Low order harmonics on the grid side current $I_{ga}$ for (**a**) SPS, (**b**) THIPS, and (**c**) SFOPS.

As presumed, by analyzing Figure 29, low order harmonics contents are present, in particular the fifth, seventh are predominant respect to the others, which anyway are under the harmonics current limits. Modulation techniques PS based have the higher values of the lower order harmonics compared with all modulation techniques previously described. Figure 30 shows a comparison among harmonic spectra centered around the switching frequency of 10 kHz among line voltage $V_{ab}$ (blue bars), converter side current $I_a$ (red bars) and grid side current $I_{ga}$ (yellow bars) obtained with SPS, THIPS, and SFOPs, respectively.

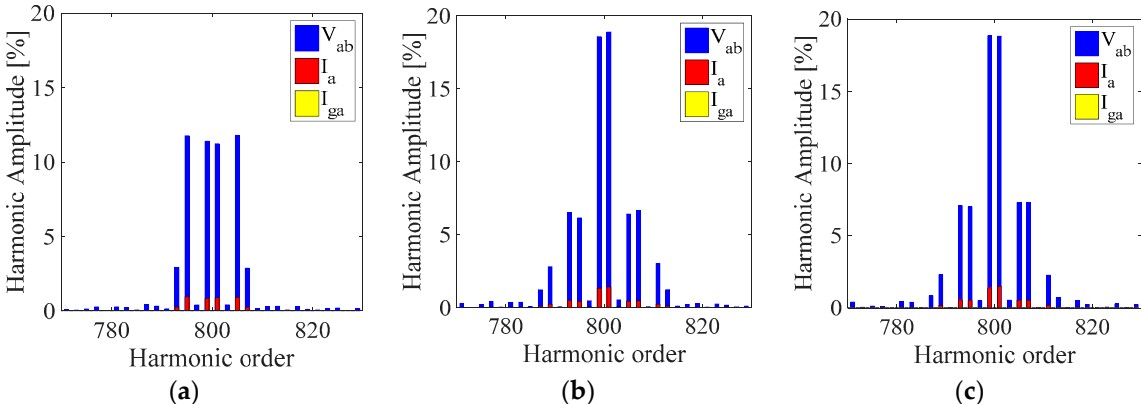

**Figure 30.** Comparison of line voltage harmonic spectra $V_{ab}$, converter side current $I_a$ and grid side current $I_{ga}$ centered around the switching frequency (10 kHz) in percent respect to the fundamental amplitude for (**a**) SPS, (**b**) THIPS, and (**c**) SFOPS.

Also for modulation techniques PS based, the grid side current harmonics are less than 0.3% referred to the fundamental amplitude demonstrating the LCL filter effectiveness. In terms of harmonic spectra of the line voltage, there are a pair of the predominant side band harmonics that are present also in the current spectra. In order to compare the performance were evaluated the *PTHD*% where the values are summarized in Table 10.

**Table 10.** *PTHD*% values obtained with SPS, THIPS, and SFOPS.

|       | $V_{ab}$ | $I_a$ | $I_{ga}$ |
|-------|----------|-------|----------|
| SPS   | 23.51%   | 1.80% | 0.0095%  |
| THIPS | 29.84%   | 2.16% | 0.0117%  |
| SFOPS | 30.51%   | 2.35% | 0.0124%  |

As shown in Table 10, the values of the *PTHD*% are lower compared with other values previously calculated. These are interesting results because on the grid side current are present only low order harmonics that are under the current limits (IEEE 1574 and IEC 61727).

In conclusion, modulation techniques PD based allow obtaining good results in terms of the harmonic content on the grid current with the lower values of the LCL filter parameters. In particular, SPD represent the best solution. Interesting results have been obtained also with modulation techniques PS based thanks to the feature that allows to shift the harmonic content respect to the switching frequency. In fact, have been obtained the lower values of the *PTHD*% about the grid side currents among all the modulation techniques taken into account. Moreover, modulation techniques PS based allow to control the power flow from the DC sources because each level can be controlled as a single-phase converter. This feature is important in very applications like PV systems for example. In the next section the experimental validation to confirm the simulation results were reported.

## 3. Experimental Validation

The purpose of this section is to provide all the useful information to describe the employed of a prototype CHBMI.

The first aim of the experimental validation is to validate the model of the system previously described and to confirm the effectiveness of the LCL filter design and the control strategy. In particular, the experimental tests have been executed with a prototype of a three-phase five-level multilevel converter with topology structure cascaded H-bridge inverter. Moreover, also the control board FPGA based is a prototype designed for power electronics applications. In this section were reported detailed descriptions of the test bench, control algorithm design and the experimental results. By the simulation

analysis, reported in section "2.4 Performances Evaluation", the experimental validation was focused on SPD and SPS modulation techniques.

*3.1. Test Bench*

In order to carry out the experimental analysis, a three-phase, five-level multilevel inverter prototype with a CHB circuital structure were assembled.

The test bench is shown in Figure 31 and it is mainly composed by:

-       a prototype of FPGA-based control board (produced by DigiPower s.r.l);
-       Six prototypes of H-bridges (produced by DigiPower s.r.l);
-       A Three-phase LCL filter especially designed (produced by SDESLAB and LEAP of the University of Palermo);
-       Six DC sources with 24 V of rated voltage;
-       Three-phase variac to grid interface;
-       A Teledyne LeCroy WaveRunner 6Zi, scope, employed for the real-time acquisition of the waveforms and monitoring of the system.

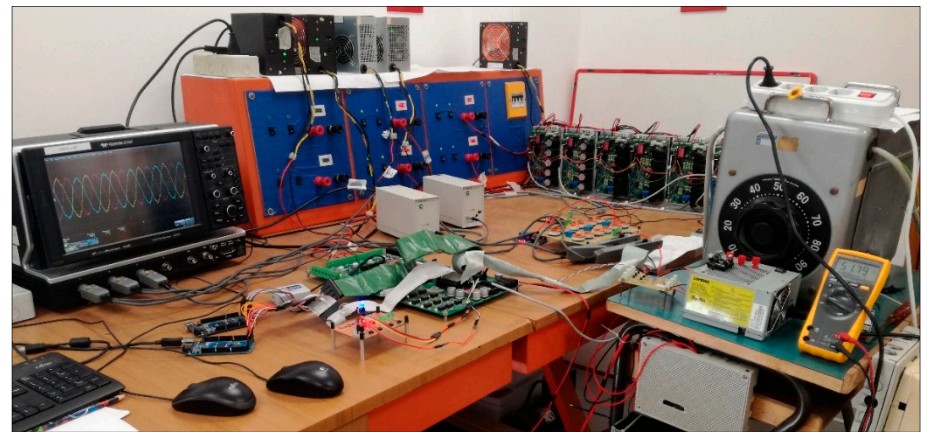

**Figure 31.** A photograph of the test bench.

Figure 32a shows the prototype of the H-Bridge that is based on power Mosfet (International Rectifier—model IRFB4115PbF [57]) whose technical features are reported in Table 11. While, Figure 32b shows the FPGA-based control board where the FPGA is produced by Altera—model Cyclone III and the features reported in [58,59].

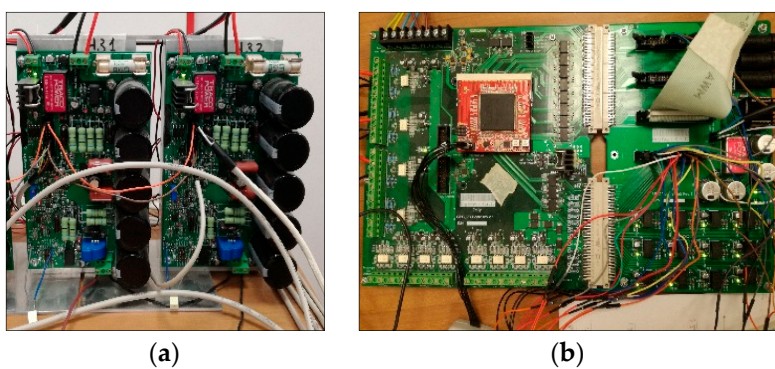

|     (**a**)     |     (**b**)     |

**Figure 32.** A photograph of the prototype (**a**) H-Brides and (**b**) field programmable gate array (FPGA) control board.

**Table 11.** Technical features of the IRFB4115PBF device [57].

| | |
|---|---|
| Voltage $V_{dss}$ | 150 V |
| Resistance $R_{ds(on)}$ | 9.3 mΩ |
| Current Id (silicon limited) | 104 A |
| Turn on delay $t_{D(on)}$ | 18 ns |
| Rise time $t_R$ | 73 ns |
| Turn off delay $T_{D(off)}$ | 41 ns |
| Fall time $t_F$ | 39 ns |
| Reversal recovery $t_{RR}$ | 86 ns |

Figure 33 shows the three-phase LCL filter especially designed and assembled with commercial components at SDESLAB and LEAP laboratories of the University of Palermo. For the converter and grid side inductance have been used commercial inductance of 560 μH with rated current of 4A in parallel connected to obtain an inductance then 280 μH with rated current of 8A.

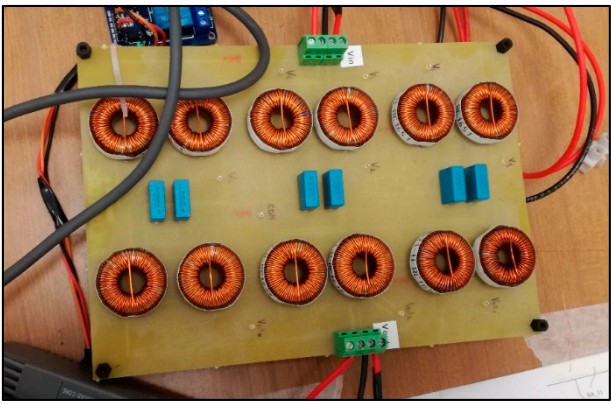

**Figure 33.** Tree-phase LCL filter.

While, for the capacitor filter have been used the ceramic capacitors of 4.7 μF with rated voltage of 100 V in parallel connected to obtain a capacitor of 9.4 μF.

*3.2. Control Algorithm Design*

The FPGA is commonly used in order to implement complex functions, such as arithmetic logic unit (ALU), memories, communication units and so on [16]. The main difference with other programmable systems used in industrial applications (μ-controller or DSP) is that through a software programming it is possible to describe an especially designed hardware for specific application. For this reason, the FPGA allows to obtain high flexibility and very fast execution time that allows using in very large of applications field. Actually, in power electronics application the complexity of the control algorithms, due also to the application type is increasing. For example, in grid connected applications there are very different control algorithm to control the power flow, power quality, synchronization with the grid, parallel control of the DC side and AC side and so on. Thus, are necessary programmable systems with fast execution time and the clock of the digital system should be adapted to the specific application. Aim of this section, is to investigate on the use of the FPGA for grid-connected application in order to validate the simulation analysis previously described and to optimize the control algorithm for the application under test. The control algorithm was implemented by means of an FPGA Altera Cyclone III EP3C40Q. The control software is Quartus II by Altera and the used programming language is the VHDL [60–62].

The structure of the control algorithm implemented can be explained by means the schematic block diagram shown in Figure 34.

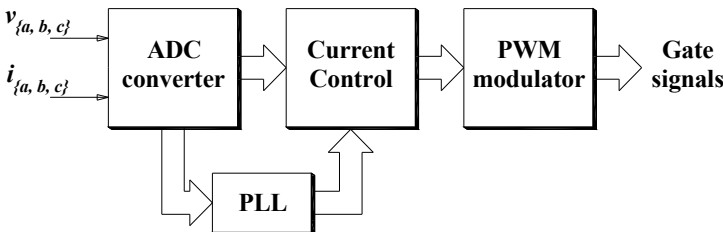

**Figure 34.** Schematic block diagram of the control algorithm.

The block named "ADC converter" represent the algorithm to manage the acquisition of the electrical quantities. In the prototype FPGA-Based control board is available an ADC converter with 16 channel (no simultaneously), 1 MSPS, 12 bit successive approximation ADC produced by Analog Devices model AD7490 16. The conversion process is managed by a clock signal reference with a frequency at 10MHz and conversion time (analog to digital signal) is equal to 2 μs. For the system under test, three voltage ($v_a$, $v_b$ and $v_c$) and three current ($i_a$, $i_b$ and $i_c$) are acquired; the conversion process is subdivided in two sub-conversion process relatively for the voltages and currents with a conversion time equal to 6 μs, respectively. In this way, the operation are executed in parallel, so when finished the first sub-conversion process relatively to the voltages, the mathematic elaborations for PLL with voltages samples start jointly with the second sub-conversion process relatively to the currents. After the conversion process, the mathematic elaboration with a resolution of 32 bit Floating Point (FP) single precision and a clock reference equal to 100 MHz starts. In Table 12 are summarized the execution time of the main mathematic operation.

**Table 12.** Execution time of the mathematic operation in FP 32bit.

| Mathematic operation | Time |
|---|---|
| Conversion Integer to Floating (Integer 13 bit, FP 32bit) | 60 ns |
| Sum or subtraction (FP 32 bit) | 140 ns |
| Product (FP 32 bit) | 110 ns |

It should be noted that between two operations there is a delay time equal to 10 ns in order to stabilize and address the digital signals.

The equivalent schematic block diagram of the PLL to describe the implementation in VHDL is shown in Figure 35.

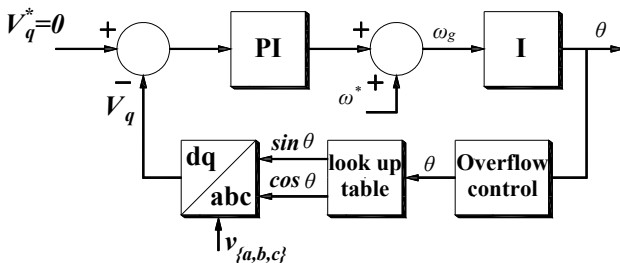

**Figure 35.** Equivalent schematic block diagram of the PLL. PI: proportional-integral; I: integral.

The PI regulator have been implemented in the discretion form as:

$$u(k) = k_p e(k) + \frac{k_p T}{T_i} e(k) + u_i(k-1) \tag{26}$$

where $k$ indicate the k-sample, $u(k)$ is the output term of the PI regulator, $e(k)$ is the error, $T$ is the sampling time, $T_i$ is the time-integral and $u_i(k-1)$ is the $k-1$ integral output. In order to optimize the execution time to evaluate the $sin\theta$ and $cos\theta$, where $\theta$ is the instantaneous phase of the space

vector voltage, a look-up table was used. Each value of the instantaneous phase was used as an address (from 0 to 6280) to determinate the values of the *sinθ* and *cosθ*. In the look-up table there are only a quarter of the sine waveform with a number of the sample equal to 1570 and it is possible to determinate the *sinθ* and *cosθ* values through a logic circuit. The block "Overflow Control" is necessary to limit the instantaneous phase value to 2π. In this way, the instantaneous phase of the space vector voltage *θ* assumes the sawtooth trend. In Table 13 are summarized the execution time of the main block of the PLL algorithm.

**Table 13.** Execution time main block of the PLL.

| Operation | Time |
|---|---|
| ABC to DQ transformation (FP 32bit) | 880 ns |
| PI regulator (FP 32 bit) | 540 ns |
| Integral (FP 32 bit) | 270 ns |
| Overflow control and look-up table (FP 32 bit) | 450 ns |

The total execution time algorithm from the end of the acquisition voltage to the instantaneous phase *θ* is equal to 3 µs. Figure 36 shows the experimental PLL effectiveness through a comparison of the grid voltage (red trend) and phase voltage of the converter (yellow trend) before carrying out the parallel with the grid.

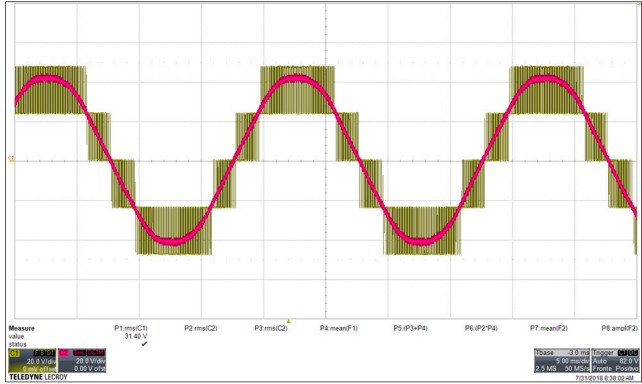

**Figure 36.** Experimental PLL effectiveness.

Figure 37 shows the execution time of the current control scheme where the total execution time is 3.13 µs.

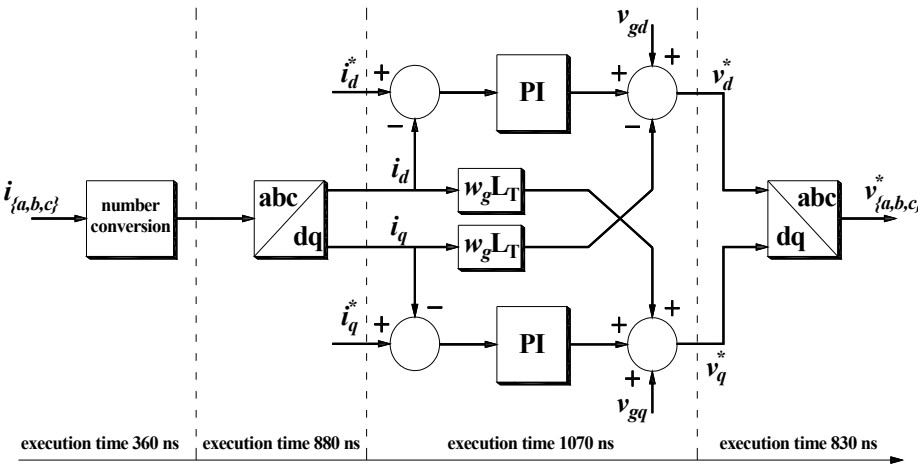

**Figure 37.** Execution time of the current control scheme.

The block "PWM generator" generates the gate signals for the converter. Generally, a carrier generator, comparator circuit and a logic circuit to generate the "dead time" compose the "PWM generator". Figure 38 shows the screenshot of the schematic block diagram of the PWM modulator implemented in Quartus II environment for SPD modulation technique.

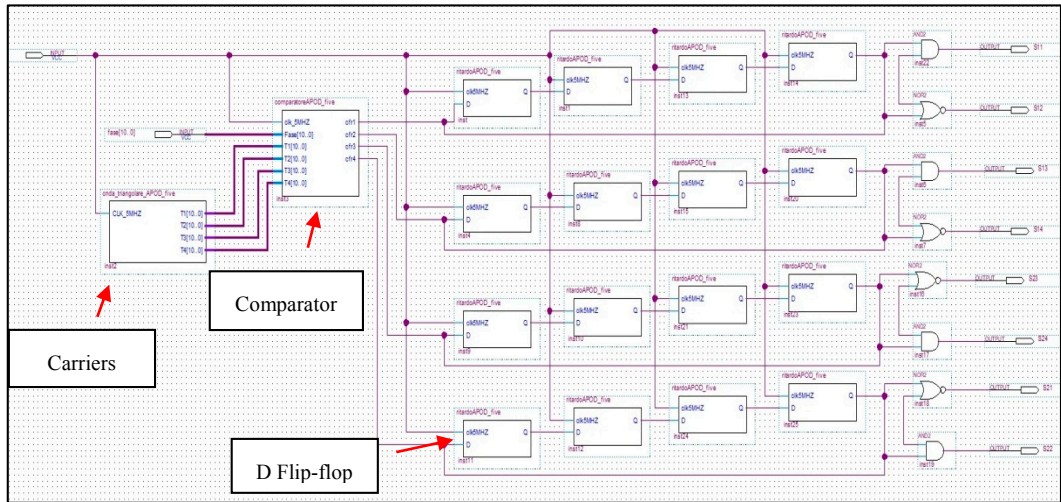

**Figure 38.** Screenshot of the schematic block diagram of the pulse width modulation (PWM) modulator implemented in Quartus II environment for SPD modulation technique.

Carrier waveform is generated by means a 13 bit up-down counter with a resolution of 1000 sample. The frequency of the clock reference is 10 MHz in order to obtain a frequency-switching equal to 10 kHz. The comparator circuit carried out the comparison between the modulating signal and carrier generator with a frequency equal to 10 MHz. The generation of the command signals of the components of the H-Bridge legs, as well as the obtainment of the dead-time for the protection of the series-connected components, is achieved through means of the logic circuit shown in Figure 39.

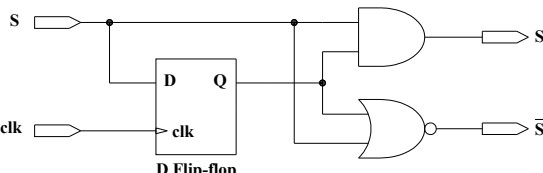

**Figure 39.** Logic circuit to generate the gate signals with dead time.

The delayed signal is obtained by using several cascaded-connected D flip-flops, whose number is dependent on the adopted clock signal. In order to obtain 400 ns of delay, four D flip-flops have been connected and managed with the 10 MHz clock signal. Figure 40 shows the simulation of the "PWM Generator" carried out in ModelSim environment relatively a phase of the converter.

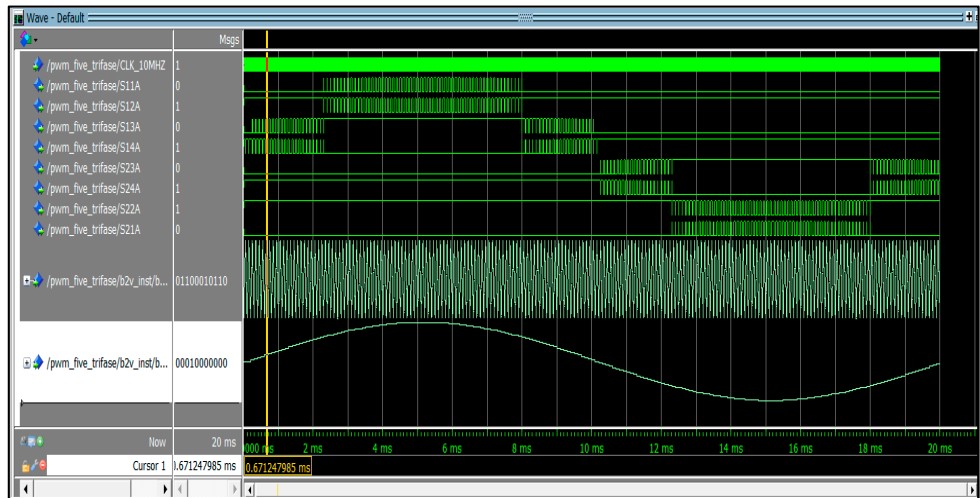

**Figure 40.** Simulation of the "PWM generator" in ModelSim environment.

From the technical features of the IRFB4115PBF reported in Table 12, the minimum dead time is equal to 100 ns, approximately. Thus, has been chosen for safe reason a dead time equal to 400 ns.

Figure 41 shows a screenshot of the experimental validation between gate signals of the same leg in order to establish the proper operation of the digital system.

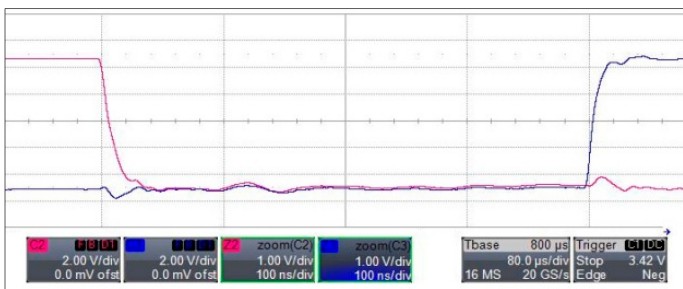

**Figure 41.** Experimental validation between gate signals of the same leg.

It should be noted that the dead time obtained is equal to 400 ns.

### 3.3. Model Validation

By the employment of the previously described test bench, the suggested techniques were experimentally implemented in order to validate the model of the system and to compare the simulation and experimental results.

The *Teledyne LeCroy WaveRunner 6Zi* acquisition system recorded the voltage waveforms. For the modulation PD, POD and APOD based techniques a sampling frequency of 50 MHz and a number of samples equal to 1 Ms were used; an observation window was choosen with a time interval equal to 20 ms.

The PS modulation techniques required an acquisition of 5 Ms of samples, equivalent to a sampling frequency of 250 MHz. The used tool to compare the simulation results and experimental results is the THD%, as reported in (27) [63]:

$$\text{THD\%} = \sqrt{\frac{V_{rms}^2 - V_{rms,1}^2}{V_{rms,1}^2}} \cdot 100 \tag{27}$$

where $V_{rms}$ is the root mean square (rms) value of the phase voltage and $V_{rms,1}$ defines the rms value of the fundamental harmonic.

Figures 42–44 show the comparison between the simulated (blue bars) and the experimental (yellow bars) THD% values obtained with Sinusoidal (Figure 42), THI (Figure 43) and SFO (Figure 44) as reference signals for each modulation techniques taken into account with the designed filter discussed in Section 2.2.

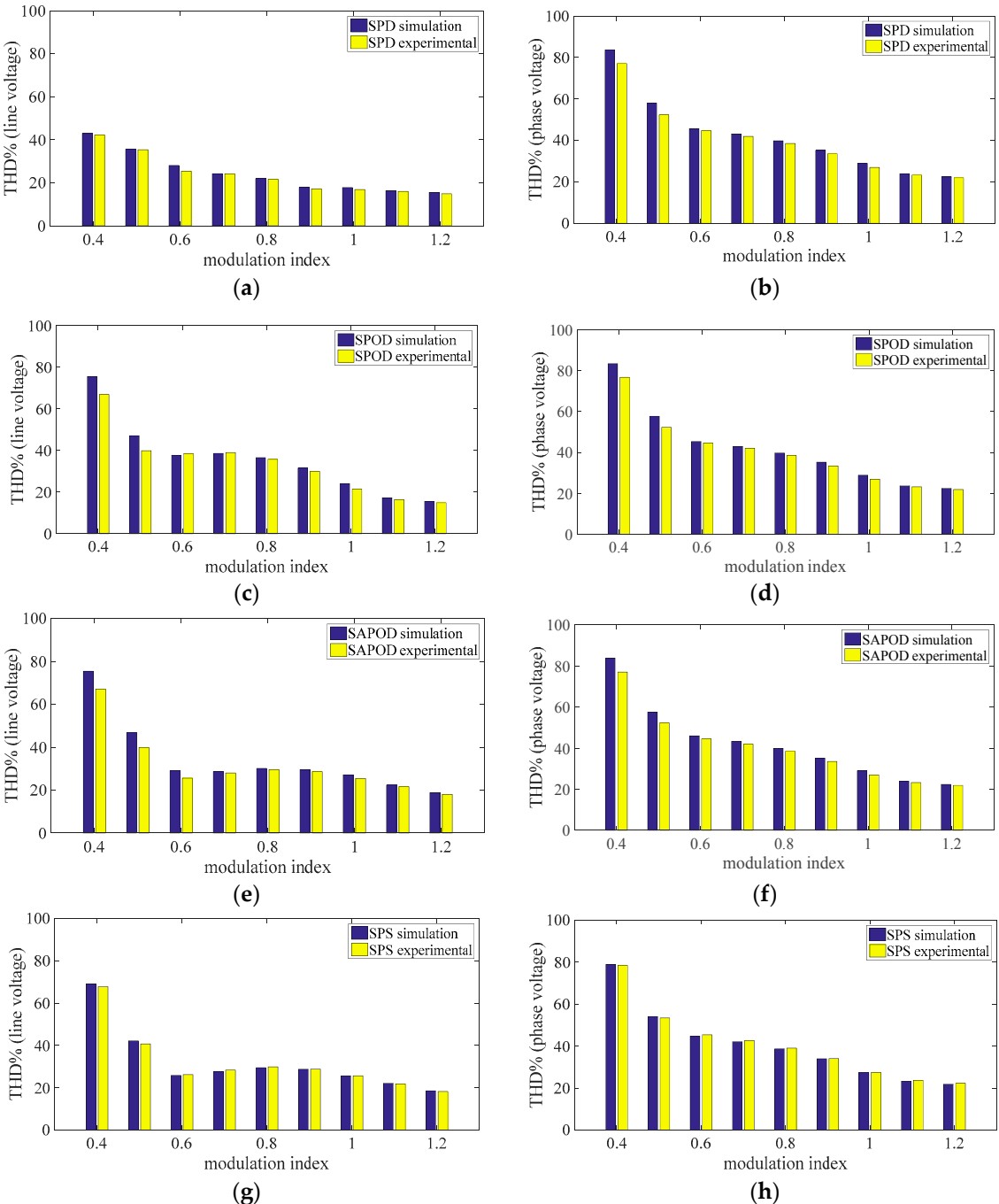

**Figure 42.** Comparison between the simulated (blue) and the experimental (yellow) THD% values: (**a**) SPD line voltage, (**b**) SPD phase voltage, (**c**) SPOD line voltage, (**d**) SPOD phase voltage, (**e**) SAPOD line voltage, (**f**) SAPOD phase voltage, (**g**) SPS line voltage, and (**h**) SPS phase voltage.

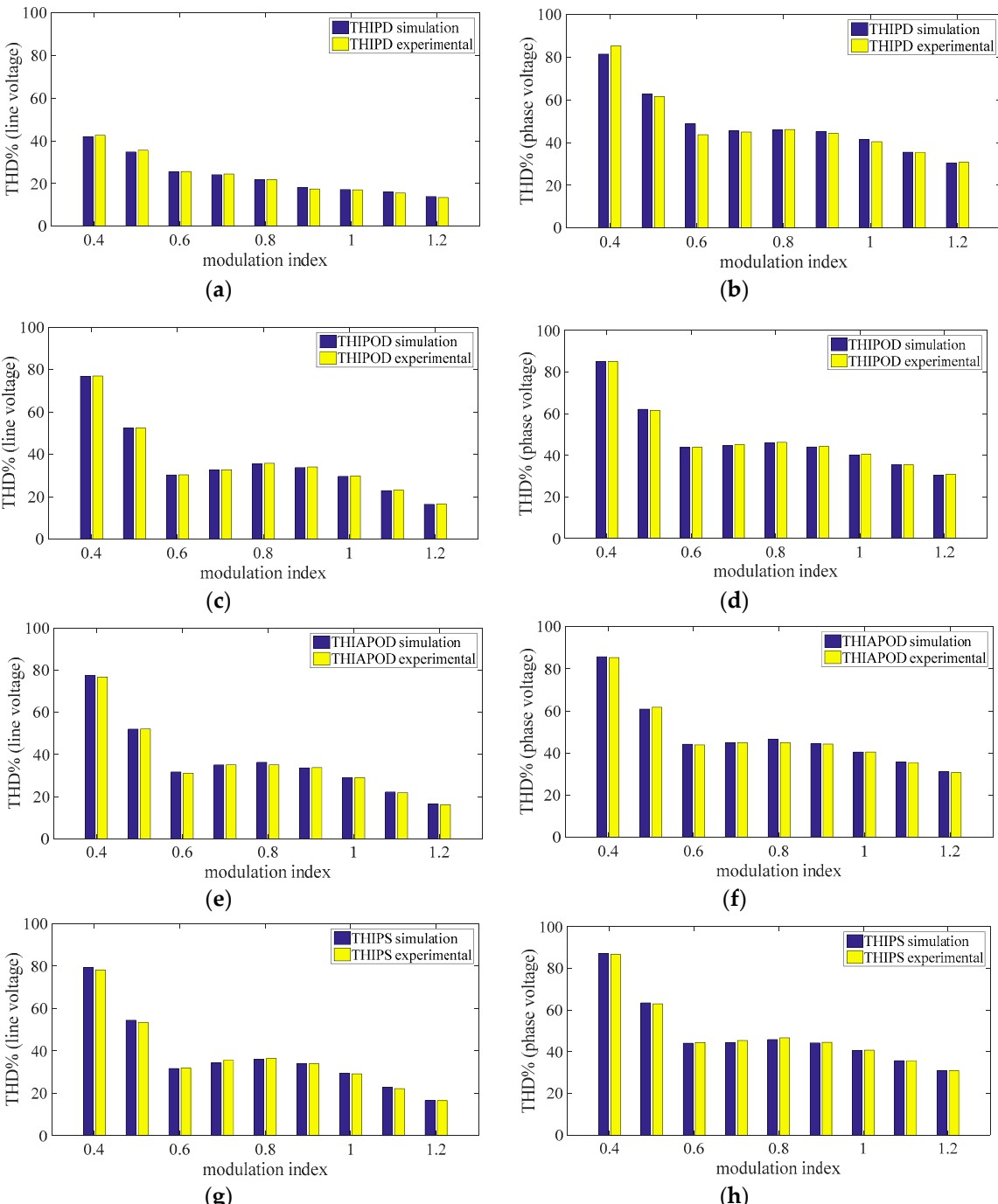

**Figure 43.** Comparison between the simulated (blue) and the experimental (yellow) THD% values: (**a**) THIPD line voltage, (**b**) THIPD phase voltage, (**c**) THIPOD line voltage, (**d**) THIPOD phase voltage, (**e**) THIAPOD line voltage, (**f**) THIAPOD phase voltage, (**g**) THIPS line voltage, and (**h**) THIPS phase voltage.

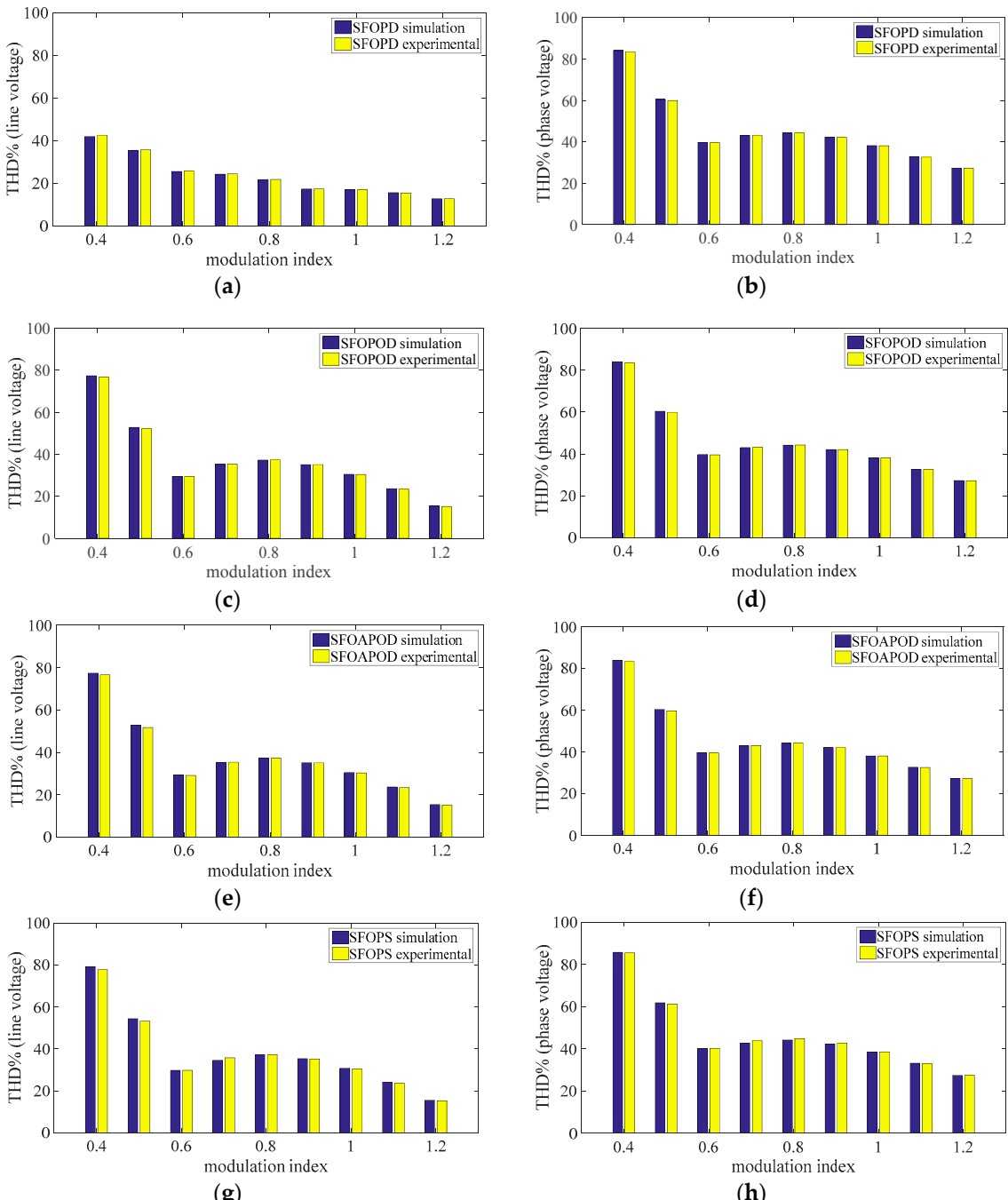

**Figure 44.** Comparison between the simulated (blue) and the experimental (yellow) THD% values: (**a**) SFOPD line voltage, (**b**) SFOPD phase voltage, (**c**) SFOPOD line voltage, (**d**) SFOPOD phase voltage, (**e**) SFOAPOD line voltage, (**f**) SFOAPOD phase voltage, (**g**) SFOPS line voltage, and (**h**) SFOPS phase voltage.

It should be noted that the simulated and experimental THD% presents similar values. For this reason, it is possible to establish the effectiveness of the model implemented.

Interesting comparison among the experimental THD% values for each reference signals taken into account, is shown in Figure 45. Modulation technique with PD as carrier signals and sinusoidal reference SPD seems to be the best solution in terms of the harmonic content. Moreover, also the SPS is a good solution for grid-connected applications.

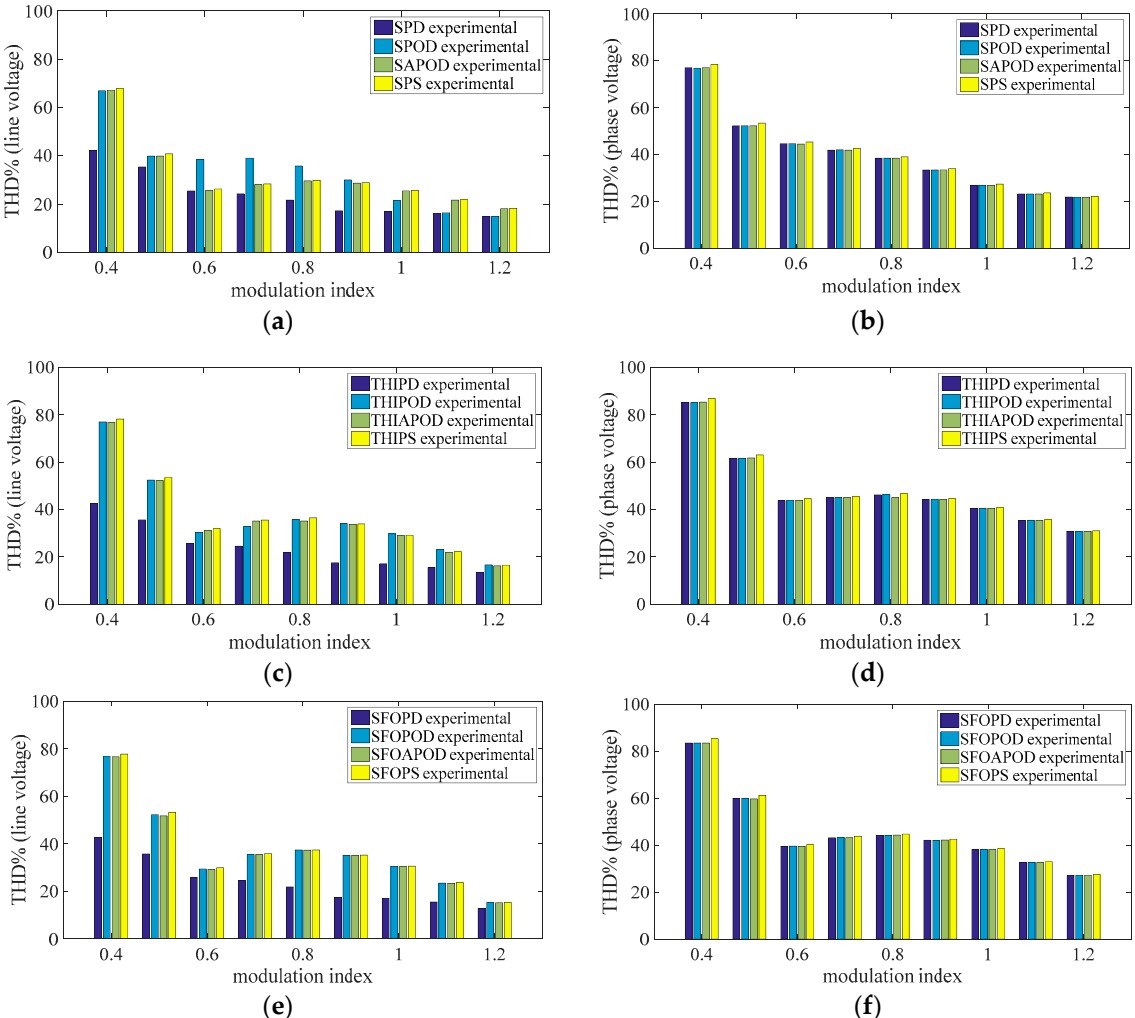

**Figure 45.** Comparison between the experimental THD% results: (**a**) Sinusoidal line voltage, (**b**) Sinusoidal phase voltage, (**c**) THI line voltage, (**d**) THI phase voltage, (**e**) SFO line voltage, and (**f**) SFO phase voltage.

In conclusion, modulation technique with PD as carrier signals and sinusoidal reference SPD present interesting results. Moreover, also the PS carrier signal is a good solution due to high order harmonic components respect other carrier signals.

In the next section, experimental validation of the grid-connected application is reported. The experimental validation considers only the SPD and SPS modulation techniques.

### 3.4. Grid Connected Application

Aim of this subsection is to validate the simulation results, reported in section "2.1 Performances evaluation", in which the best performances were obtained with SPD and SPS modulation techniques. In particular, the purpose is to validate by means experimental tests the effectiveness of the control strategy and the LCL filter. Thus, the experimental tests were carried out only with SPD and SPS modulation techniques.

#### 3.4.1. Phase Disposition

Figure 46 shows the measured grid phase voltages and grid currents of the phase *a* and *b* obtained with SPD modulation technique at the rated power. It is interesting to note that the phase angle between voltage and current of the same phase is equal to zero. This result demonstrates the effectiveness of the control strategy because, as explained in the section "2.3 Controller Design", through the *d* component

it is possible to control the active power while through the *q* component it is possible to control the reactive power. Thus, fixing *q* component of the current equal to zero and *d* component of the current equal to rated value (6A) it is possible to inject only active power on the grid as shown in Figure 46.

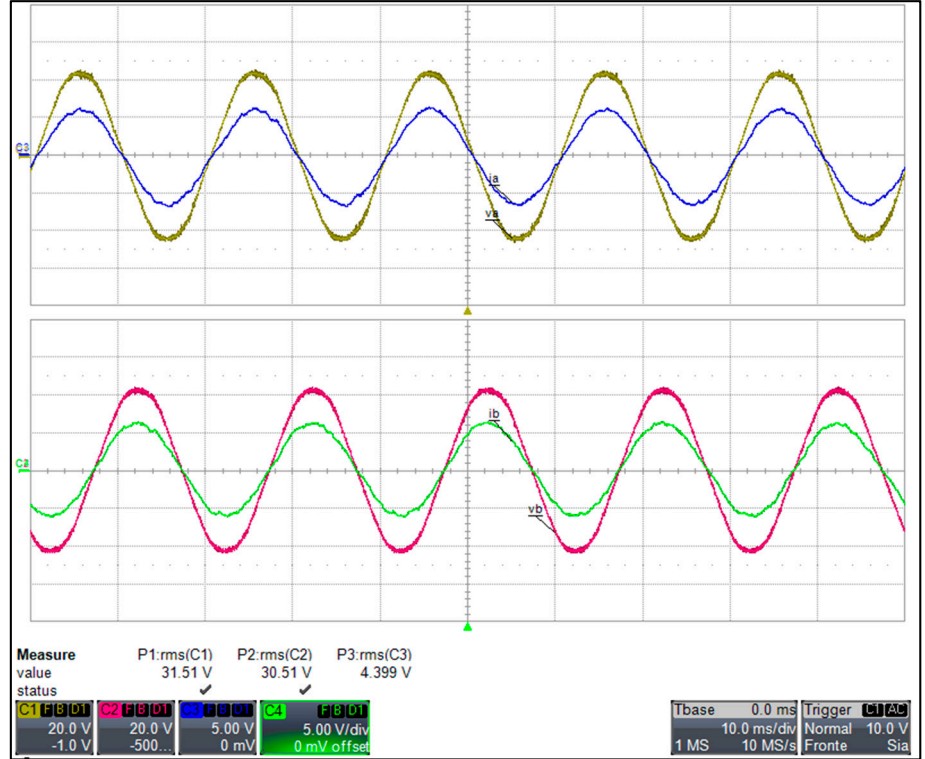

**Figure 46.** Measured grid voltages (20 V/div) and grid currents (5 A/div) of the phase a and b obtained with SPD at the rated power.

Figure 47 shows measured converter side current obtained with SPD at rated power while Figure 48 shows the measured grid side current in the same conditions.

First all, the differences in terms of the harmonic content between the trend of the converter side and grid side currents are evident. Moreover, a not perfect half-wave symmetry in the currents trend was observed. This phenomenon determined the present of the even-harmonics in the current.

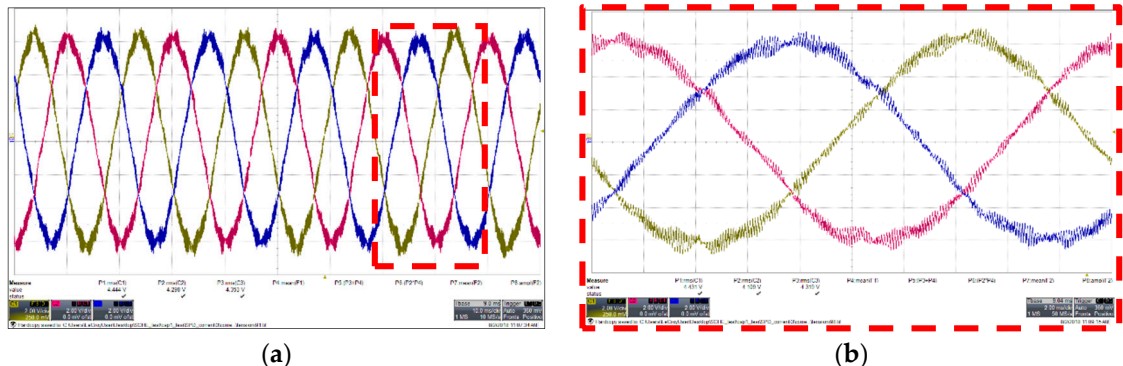

(**a**)                    (**b**)

**Figure 47.** Measured converter side currents (2 A/div) obtained with SPD at the rated power. (**a**) Ripple in different cycles; (**b**) Magnification of ripple.

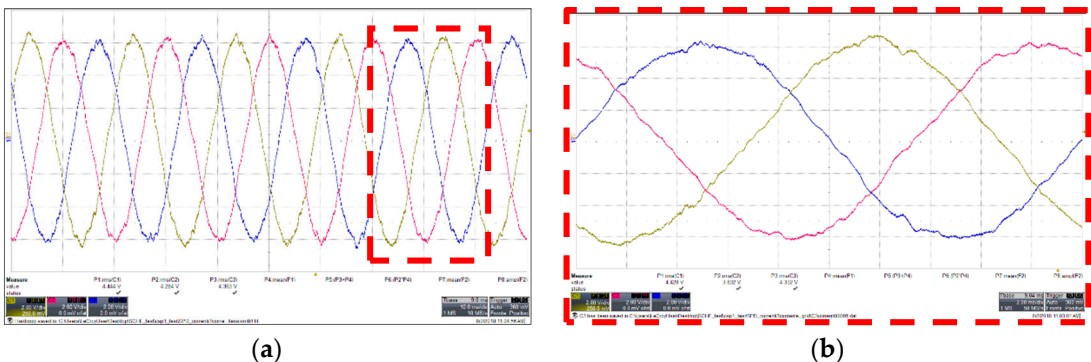

**Figure 48.** Measured grid side currents (2 A/div) obtained with SPD at the rated power. (**a**) Ripple in different cycles; (**b**) Magnification of ripple.

Figure 49a shows the low order harmonics spectra of the grid side current at rated power. In the first all, it interesting to note that the amplitude of the lower order harmonics are below of the standard harmonic current limits defined by IEEE 1574 and IEC 61727 at the PCC. Nevertheless, as stated above are present the even-harmonics on the harmonic spectra. Interesting comparison between the harmonic spectra centered on switching frequency of the converter side current $I_a$ (blue bars) and grid side current $I_{ga}$ (yellow bars) is shown in Figure 49b. The lower values of the harmonics of the grid side current demonstrate the effectiveness of the LCL filter.

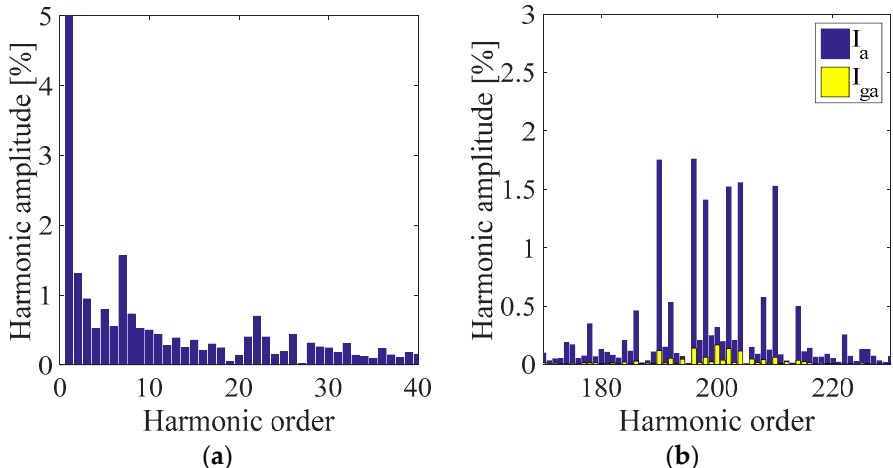

**Figure 49.** Calculated (**a**) low order harmonics of the grid side current and (**b**) switching frequency harmonics spectra of the converter side and grid side currents.

In Table 14 are summarized the calculated THD% for different values of the grid side current injected. It should be noted that the THD% values increase when the current injected in the grid is reduced and it is less then 5% up to $I_n/2$.

**Table 14.** Experimental THD% of the converter and grid side currents, obtained with SPD, for different values of the injected current into the grid.

|  | $I_n/3$ | $I_n/2$ | $2I_n/3$ | $I_n$ |
| --- | --- | --- | --- | --- |
| Converter side current | 12.17% | 7.82% | 6.46% | 5.88% |
| Grid side current | 7.97% | 4.78% | 4.26% | 3.72% |

Figure 50 shows the measured line voltage of the converter at rated power. It interesting to note that the line voltage presents nine-level.

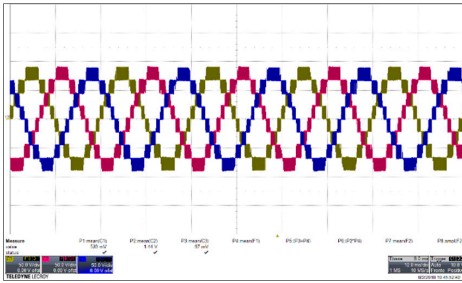

**Figure 50.** Measured line voltage of the converter at rated power.

Figure 51 shows the measured capacitor voltage of the LCL filter at rated power. The evident low harmonic content in the trend of the capacitor voltage demonstrate the efficacy of the LCL filter.

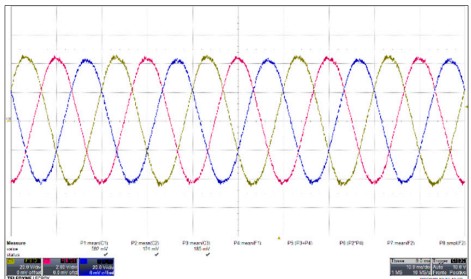

**Figure 51.** Measured capacitor voltage of the LCL filter at rated power.

As stated earlier, the second experimental tests have been carried out with SPS modulation techniques with the same filter used for SPD modulation techniques thank to the similar values obtained in the subsection "LCL filter Designs".

### 3.4.2. Phase Shifted

Figure 52 shows the measured grid phase voltages and grid currents obtained with SPS modulation technique at the rated power for each phase of the system.

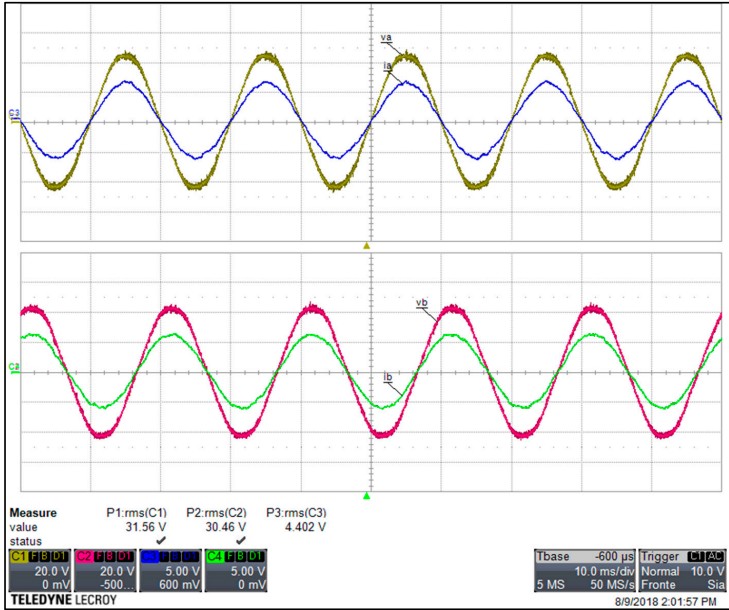

**Figure 52.** Measured grid voltages (20 V/div) and grid currents (5 A/div) of the phase a and b obtained with SPS at the rated power.

Also for this case, the phase angle between voltage and current of the same phase is equal to zero, thus this result demonstrates the efficacy of the control strategy.

Figures 53 and 54 show the measured converter side and grid side currents, respectively. As mentioned above, the currents trends present a not perfect half-wave symmetry and this phenomenon determined the even harmonics.

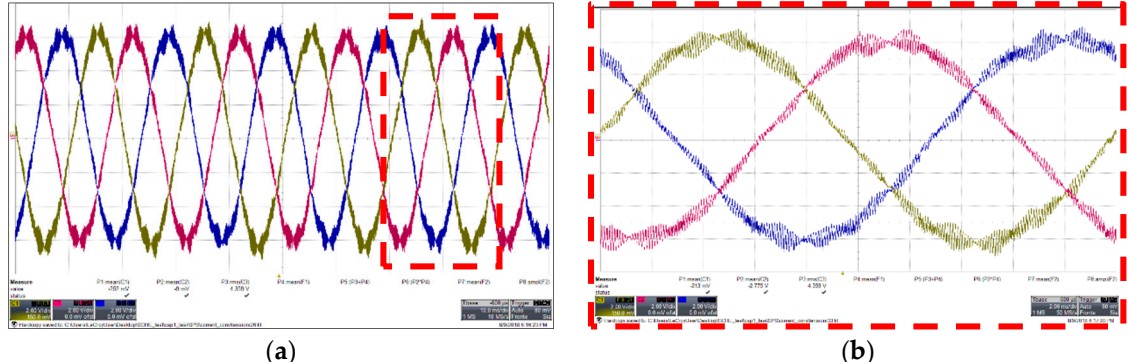

**Figure 53.** Measured converter side currents (2 A/div) obtained with SPS at the rated power. (**a**) Ripple in different cycles; (**b**) Magnification of ripple.

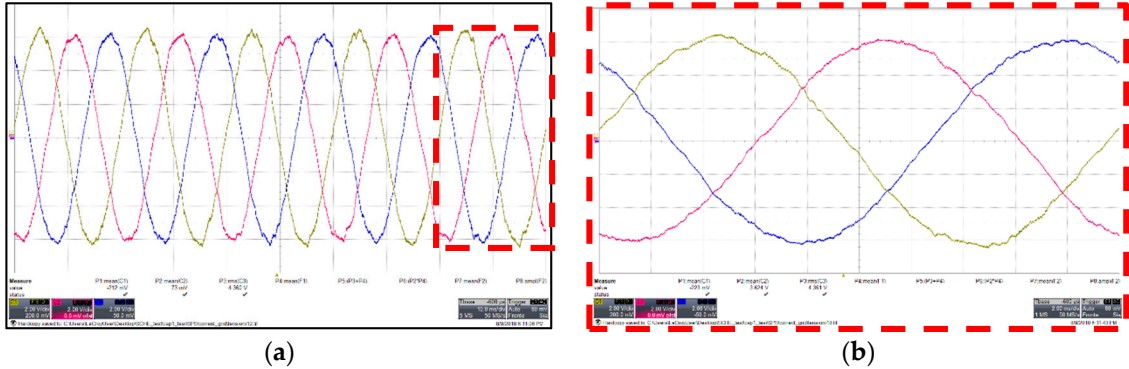

**Figure 54.** Measured grid side currents (2 A/div) obtained with SPS at the rated power. (**a**) Ripple in different cycles; (**b**) Magnification of ripple.

Figure 55a shows the low order harmonics of the grid current at rated power. The amplitude of the all low order harmonics are less of the current limits reported in Table 1. However, by comparing the low order harmonic spectra of SPD and SPS, it can be noted that the second order harmonic is higher in SPS modulation technique. The presence of the even harmonics also is due to absence of the DC voltage control. Moreover, in both low order harmonic spectra of the SPD and SPS is predominant a seventh harmonic with similar value.

In Figure 55b is shown the comparison between the harmonic spectra centered on switching frequency of the converter side current $I_a$ (blue bars) and grid side current $I_{ga}$ (yellow bars). The lower values of the harmonics of the grid side current demonstrate the effectiveness of the LCL filter.

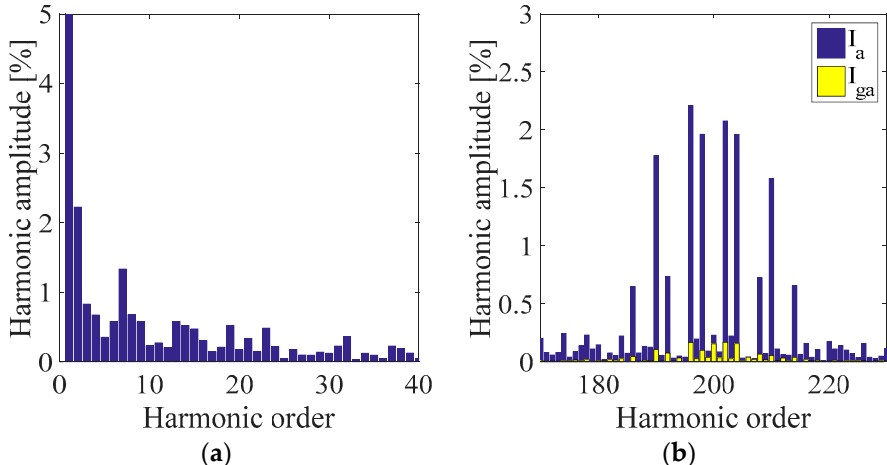

**Figure 55.** Calculated (**a**) low order harmonics of the grid side current and (**b**) switching frequency harmonics spectra of the converter side and grid side currents.

In Table 15 are summarized the calculated THD% for different values of side current injected in the grid. The THD% values obtained with SPS are similar respect to the previously calculated with SPD. This is an interesting result, because the modulation techniques PS based are more versatile respect to the others multicarrier modulation techniques for grid connected applications like PV systems, for example. The modulation techniques PS based allow to control each H-Bridge like a single-phase inverter and it is possible to use innovative control algorithms especially designed in dependence of the application.

**Table 15.** Experimental THD% of the converter and grid side currents, obtained with SPS, for different values of the injected current into the grid.

|  | $I_n/3$ | $I_n/2$ | $2I_n/3$ | $I_n$ |
|---|---|---|---|---|
| Converter side current | 12.28% | 8.41% | 6.80% | 5.64% |
| Grid side current | 7.42% | 4.45% | 3.91% | 3.33% |

In addition, the line voltage build with the SPS modulation technique has nine level, as shown in Figure 56.

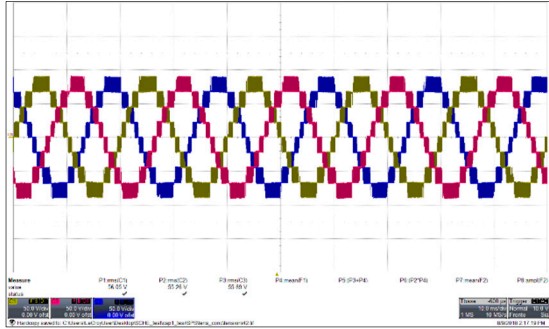

**Figure 56.** Measured line voltage of the converter at rated power.

Figure 57 shows the measured capacitor voltage of the LCL filter with an evident low harmonic content that also in this case demonstrate the efficacy of the LCL filter.

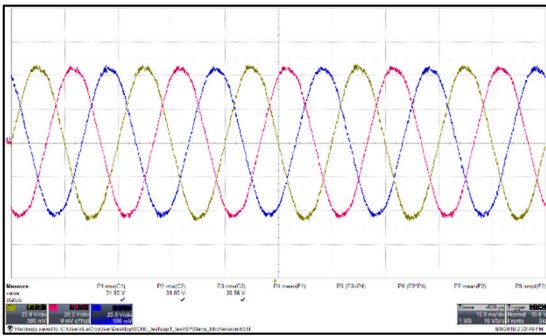

**Figure 57.** Measured capacitor voltage of the LCL filter at rated power.

## 4. Discussion

In order to face the harmonic distortion problem, two issue can be distinctly taken into account: the generation of harmonics and their suppression. Although the approach is not purely dichotomous, since a lower generation corresponds to an easier suppression, here the main results of this work can be approached with an etiological methodology.

The modulation techniques with PD as carrier signals shows a harmonic spectrum of the phase voltage with a predominant harmonic centered on the switching frequency and side band harmonics (Figure 10). By considering the modulation with POD and APOD as carriers, in the spectra, the harmonic component at switching frequency does not appear but there are only side bands (Figure 16). By considering the modulation with PS disposition, the harmonics are centered around four times the switching frequency, are present only side bands harmonics like in modulation techniques POD and APOD based (Figure 25).

In order to reduce the harmonics in the grid side, a filtering system is correctly designed. For the PD based modulation techniques, the converter side inductance $L$ and grid side inductance $L_g$ present the lower values (Table 4). Higher value of the converter side inductance were obtained with THIPOD modulation technique, phenomenon attributable to the higher number of the side band harmonics generated by POD carrier signals. Figure 14 (PD), 23 (POD and APOD) and 29 (PS) show the performances of the filtering by considering the different modulation techniques. For each techniques, the third harmonic of injected current is much reduced, so the comparison moves on the fifth harmonic: PD is around 1%, POD and APOD less than 0.5%, PS around 2%. Excellent performance of THIPOD for its very low values of fifth and also seventh harmonic, are remarkable, but the side inductance $L$ is eight times the value for PD ones. By analyzing Figure 29, low order harmonics contents are present, in particular besides the fifth, seventh is predominant. Modulation techniques PS based have the higher values of the lower order harmonics compared with all modulation techniques previously described.

In conclusion, modulation techniques PD based allow obtaining good results in terms of the harmonic content on the grid current with the lower values of the LCL filter parameters. In particular, SPD represent the best solution.

The previous described good results are validated in Figure 45 with the experimental test phase; moreover, it is possible to find an experimental behavior better than the simulated one for SPD for different modulation indexes.

Finally, the approach was validated for a grid-connected system by exploiting the three-phase Variac to grid interface. Different current values were injected in the power grid, Tables 14 and 15 report that the THD% remained below the 4%, 3.72% for SPD and 3.33% for SPS techniques.

By considering the work of Colak et al. [64], as the number of levels in multilevel inverter increases, the THD in the output voltage reduces, but the drawback of increasing the levels is that that the control circuit becomes hard challenges. A comparison can be done with recent results found in literature. Kavali and Mittal obtained by MATLAB based simulation with SIMULINK environment interesting results for their single-phase five level CHBMI topology with sinusoidal pulse width modulation schemes [37]. For PD the THD was reduced to 5.69%; in case of POD it was 5.75%; for APOD it was

5.73%; the THD was 7.42% in case of PS. Results obtained in the present work follow those obtained in [37], the obtained THDs are below, and add an experimental validation to them.

## 5. Conclusions

As previously described, the PWM modulation techniques found large use in many industrial applications thanks their main features as easy implementation in electronic control systems, low computational cost and high flexibility. Moreover, for grid-connected applications the PWM modulation techniques are the best solution due to the lower amplitude of the low-order harmonics, reducing the filter requirements. Thus, in this work a detailed analysis taking into account all PWM modulation techniques, the LCL filter requirements and the real time implementation issues in FPGA-based prototype control board for a grid-connected three-phase five-level CHBMI, was presented.

Firstly, through a simple step-by-step procedure to design LCL filter for each modulation techniques taken into account, it was demonstrated that the lower values of the filter parameters are obtained for modulation techniques employing sinusoidal as reference signals. These interesting results were confirmed by the experimental validation of the THD% values. In particular, the SPD and SPS showed the best results in terms of the THD% values. Then, the experimental tests was focused by using the SPD and SPS modulation techniques in order to inject in the power grid different values of current through the specially designed LCL filter. Notably, the experimental tests confirmed the effectiveness of the LCL filter. The amplitude of the lower order harmonics are below of the standard harmonic current limits at the point of common coupling. Nevertheless, appeared even-harmonics on the amplitude spectra. Finally, it is possible to claim that the modulation technique SPS is the best solution for all grid-connected applications where it is necessary to control the power flow of the DC sources separately.

**Funding:** This research received no external funding.

**Acknowledgments:** This work was financially supported by MIUR-Ministero dell'Istruzione dell'Università e della Ricerca (Italian Ministry of Education, University and Research), by SDESLab (Sustainable Development and Energy Saving Laboratory), and LEAP (Laboratory of Electrical Applications) of the University of Palermo.

**Conflicts of Interest:** The author declares no conflict of interest.

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
