# Peer review of "Experimental Evaluation of the Performance of a Three-Phase Five-Level Cascaded H-Bridge Inverter by Means FPGA-Based Control Board for Grid Connected Applications"

_energies, doi:10.3390/en11123298_

Round 1
Reviewer 1 Report
This paper presents theoretical and experimental results for simulation and practical validation of a three-phase five-level cascaded H-bridge inverter control for grid connected applications.
The paper is a review of existing techniques and technology applied for multilevel inverters, with an accent on Cascaded H-Bridge Multilevel Inverter. The main contributions of the author consists in synthesizing information on design practices and simulating the operation of the CHBMI for grid connected applications. As the reviewer could found, the author's other original contributions are missing or minimal.
Beyond that remark, the work is well written and well structured. The results presented in the experimental section of the paper support the theoretical model developed and the simulations presented in the previous sections.
However, the author must take into consideration other issues (English language, typos and other errors), such as:
- Page 1, line 32: use “… structures proposed …” instead of “… structures and proposed …”.
- Page 1, line 35: use the singular form: "structure".
- Page 1, line 36: use "... four capacitors connected in series ..." or "... four series connected capacitors ...".
- Page 2-3, lines 76-78: sentence "It should be noted that ... types of renewable energy (PV, wind farm, fuel cell, etc)" must be rephrased.
- Page 4, line 128: add square brackets for addressing references: use [9] instead of 9.
- Page 7, lines 166-167: the sentence “In addition the Power Converter and filter, it is necessary a control system supervising correctly working” must be rephrased.
- Page 7, lines 176-177: the sentence “Digital controller represents the core of the system and it is possible use different available types in the market” must be rephrased.
Author Response
Reviewer 1:
This paper presents theoretical and experimental results for simulation and practical validation of a three-phase five-level cascaded H-bridge inverter control for grid connected applications.
The paper is a review of existing techniques and technology applied for multilevel inverters, with an accent on Cascaded H-Bridge Multilevel Inverter. The main contributions of the author consists in synthesizing information on design practices and simulating the operation of the CHBMI for grid connected applications. As the reviewer could found, the author's other original contributions are missing or minimal.
Beyond that remark, the work is well written and well structured. The results presented in the experimental section of the paper support the theoretical model developed and the simulations presented in the previous sections.
Reply: First of all, I want to thank the reviewer for his time devoted to the study of this long paper.
The implementation of control technique in a FPGA and the grid-connection of system are novelty elements for the Cascaded H-Bridge, which often finds only simulations in scientific articles.
However, the author must take into consideration other issues (English language, typos and other errors), such as:
Reply: Dear Reviewer, all suggestion were followed, thank you so much for your help. New sentences and correction are high lined in yellow.
- Page 1, line 32: use “… structures proposed …” instead of “… structures and proposed …”.
- Page 1, line 35: use the singular form: "structure".
- Page 1, line 36: use "... four capacitors connected in series ..." or "... four series connected capacitors ...".
- Page 2-3, lines 76-78: sentence "It should be noted that ... types of renewable energy (PV, wind farm, fuel cell, etc)" must be rephrased.
- Page 4, line 128: add square brackets for addressing references: use [9] instead of 9.
- Page 7, lines 166-167: the sentence “In addition the Power Converter and filter, it is necessary a control system supervising correctly working” must be rephrased.
- Page 7, lines 176-177: the sentence “Digital controller represents the core of the system and it is possible use different available types in the market” must be rephrased.
Reviewer 2 Report
The article concerns a very interesting and actual topic of evaluation of the performance of a three-phase five-level cascaded H-bridge by means FPGA based control for grid-connected applications. The author provided very detailed analysis of conducted research and experimental results.
General remarks:
1. The introduction is a bit too long. There should be a brief literature review and paper contribution. The intro should not consist subsections. The Author should consider moving some parts to the Appendix. The whole part about converter technologies subsection 1.1 – 1.2 to could be the separate section.
2. There should be wider vertical spaces after the tables in the whole paper.
3. There are many blurry figures especially those from figure 10. Reading the legends, axis titles or text is hard. The Author should insert the figures in better resolution.
4. I suggest that Author could move some figures to the appendix (e.g., 36,38,40,41,50,51) and focus more on the strict contribution rather than on the analysis and comparisons.
Some of the detailed remarks:
Lines:
340: unnecessary ‘2’;
408: Table 6,7 – missing spaces between variables and units;
431: Low quality of figure 10;
443: Figures 12,23 are split into two pages;
548: missing space;

Author Response
Review 2
The article concerns a very interesting and actual topic of evaluation of the performance of a three-phase five-level cascaded H-bridge by means FPGA based control for grid-connected applications. The author provided very detailed analysis of conducted research and experimental results.
Reply: First of all, I want to thank the reviewer for his time devoted to the study of this long paper. The words of the reviewer are the best reward for a job that lasted months.
General remarks:
1.The introduction is a bit too long. There should be a brief literature review and paper contribution. The intro should not consist subsections. The Author should consider moving some parts to the Appendix. The whole part about converter technologies subsection 1.1 – 1.2 to could be the separate section.
Reply: Dear Reviewer, I tried to separate this session, the original work was about 60 pages, had a subsection also for the grid-connected applications; I removed it in order to have less pages, but I’m unable to reduce again the introduction or move portions in the appendix, since figures 2-6, are fundamental for the following parts. The only subsection that I can put in the appendix is 1.2, but the paper would lose its focus on FPGA programming, which is at the heart of the successful implementation of techniques that allow for harmonic reduction.
2. There should be wider vertical spaces after the tables in the whole paper.
Reply: Dear Reviewer, suggestion is right, I fixed all tables.
3.There are many blurry figures especially those from figure 10. Reading the legends, axis titles or text is hard. The Author should insert the figures in better resolution.
Reply: Suggestion has been followed, figures now are bigger. The problem of the low resolution is due to pdf file, which was compressed, word and so the final paper has not this problem.
4. I suggest that Author could move some figures to the appendix (e.g., 36,38,40,41,50,51) and focus more on the strict contribution rather than on the analysis and comparisons.
Reply: Dear Reviewer, figures suggested to be moved are the ones that are useful for the implementation of the FPGA control. This paper has been written by considering different pillars, one of these is the implementation of control technique, part that required more the half time devoted to the realization of the test bench. I would like to keep section 1.2, 2.3 and 3.2.
Some of the detailed remarks:
Lines:
340: unnecessary ‘2’;
Reply: Suggestion has been followed
408: Table 6,7 – missing spaces between variables and units;
Reply: Suggestion has been followed
431: Low quality of figure 10;
Reply: Suggestion has been followed, dimensions have been enlarged
443: Figures 12,23 are split into two pages;
Reply: thank you, now figures are fixed.
548: missing space;
Reply: thank you, it is fixed.

Reviewer 3 Report
The author evaluates the performance of a grid connected three-phase five-level cascaded H-bridge with experimental results. In particular, 4 different PWM multicarrier modulation techniques and 3 distinct modulation signals have been compared for the selected converter in terms of current and voltage THD. The paper is well structured and clearly expose the comparison. Simulation and experimental results are also provided. According to the reviewer, it is an interesting paper. However, the following comments should be addressed to improve the overall quality of the paper.
Major comments:
- The differences between the presented paper and ref. [10] should be more emphasized
- the LCL filter is designed in order to reduce the output harmonics. Then, a step-by-step design procedure has been adopted for the 12 cases presented. This of course leads to different filter parameters. In section 2 the performance has been evaluated in terms of PTHD and different PTHD values have been achieved. How does the author know that these differences on the PTHD are related to the modulation techniques and not to the different filter parameters? For example, in Table 8 THIPD has the higher PTHD in comparison with the SPD and SFOPD, but at the same time it also has the lowest filter inductance. In table 9, THIPOD has a very low PTHD that it is probably due to the high value of L.
Minor comments:
- Some typo errors are present in the paper. For example:
Page 2, line 65: “[…] and can be controlled as […]” instead of “[…] and can be control as […]”.
Page 9, line 259: “[…] a soft switching modulation is used with high switching PWM […]” instead of “[…] a soft switching modulation is used for with switching PWM […]”.
Page 10, line 293: “[...] the AC and DC side can be totally described […]” instead of “[…] the AC and DC side be totally described […]”.
Page 11, line 312: “[…] it is necessary to develop an average model […]” instead of “[…] it is necessary to development an average model […]”.
Page 11, line 313: “[…] the average model takes into account […]” instead of “[…] the average model taken into account […]”.
Page 11, line 314: “[…] the average phase voltages and average currents […]” instead of “[…] the average phase voltages and average current […]”.
And so on. Please read again carefully the paper.
- In equation (15) r and va* are not defined. It is probably not the same r used as index in Table 3, right? Is va* the voltage on Cf?
- rg in equation (16) is not defined.
- In (18), why is the voltage on Cf not considered?
- Page 12, line 331: reference [51] has been mentioned twice.
- Page 13, line 354: Table 3 instead of Table 3.3. The same for Table 5 in Page 4 line 367.
- Page 14, line 378: “According to [54] and [55], […]” instead of “According to 54 and 55, […]”
- Page 14, line 379: Figure 9 instead of Fig9.
- In the FPGA implementation of the control, floating point representation has been chosen instead of the fixed-point representation. Could the author comment on this choice? With FP the precision is improved but the required resources are higher.
- Concerning the comparisons presented in Figure 42-43-44-45, have they been achieved by using different filter parameters in the experimental setup?
Author Response
The author evaluates the performance of a grid connected three-phase five-level cascaded H-bridge with experimental results. In particular, 4 different PWM multicarrier modulation techniques and 3 distinct modulation signals have been compared for the selected converter in terms of current and voltage THD. The paper is well structured and clearly expose the comparison. Simulation and experimental results are also provided. According to the reviewer, it is an interesting paper.
Reply: First of all, I want to thank the reviewer for his time devoted to the study of this long paper. The words of the reviewer are the best reward for a job that lasted months.
However, the following comments should be addressed to improve the overall quality of the paper.
Major comments:
- The differences between the presented paper and ref. [10] should be more emphasized
Reply: According this suggestion lines 149-151 were modified into: “An interesting deep discussion on the previous proposed techniques can be found in [10], in which some features of the proposed technique can be found without the control issue and filter design;”.
Paper [10] presents a similar study on THD, but the control implementation and the filter design are missing, so the old paper gains a partial success.
- the LCL filter is designed in order to reduce the output harmonics. Then, a step-by-step design procedure has been adopted for the 12 cases presented. This of course leads to different filter parameters. In section 2 the performance has been evaluated in terms of PTHD and different PTHD values have been achieved. How does the author know that these differences on the PTHD are related to the modulation techniques and not to the different filter parameters? For example, in Table 8 THIPD has the higher PTHD in comparison with the SPD and SFOPD, but at the same time it also has the lowest filter inductance. In table 9, THIPOD has a very low PTHD that it is probably due to the high value of L.
Reply: Dear Reviewer, I understand this point of view. I think that both aspects concur to define the final harmonic content. However, in the paper I followed this scheme:
-the step-by-step procedure was used to design the LCL filter, it is a numerical method based on the harmonic content in the line voltage and on the power inject in the grid.
- the PTHD% was evaluated to obtain the benefits introduced with the LCL filter around the switching frequency.
If the PTHD% was used to define the LCL parameters, the problem suggested by the Reviewer should be an error in the work.
Minor comments:
- Some typo errors are present in the paper. For example:
Page 2, line 65: “[…] and can be controlled as […]” instead of “[…] and can be control as […]”.
Page 9, line 259: “[…] a soft switching modulation is used with high switching PWM […]” instead of “[…] a soft switching modulation is used for with switching PWM […]”.
Page 10, line 293: “[...] the AC and DC side can be totally described […]” instead of “[…] the AC and DC side be totally described […]”.
Page 11, line 312: “[…] it is necessary to develop an average model […]” instead of “[…] it is necessary to development an average model […]”.
Page 11, line 313: “[…] the average model takes into account […]” instead of “[…] the average model taken into account […]”.
Page 11, line 314: “[…] the average phase voltages and average currents […]” instead of “[…] the average phase voltages and average current […]”.
And so on. Please read again carefully the paper.
Reply: Thank you, according this suggestion, whole paper was revised.
- In equation (15) r and va* are not defined. It is probably not the same r used as index in Table 3, right? Is va* the voltage on Cf?
- rg in equation (16) is not defined.
Reply: lines 307-309 were changed into “Finally, the following equations can be used for the rating the output currents i{a,b,c} of the converter (15), grid currents ig{a,b,c} (16) and the capacitor voltages v*{a,b,c} (17), where r and rg are the resistance of the inductance L and Lg of LCL filter.”
- In (18), why is the voltage on Cf not considered?
Reply: it is considered by the sum of (va*+ vb*+ vc*)
- Page 12, line 331: reference [51] has been mentioned twice.
Reply: changed into [51] and [52].
- Page 13, line 354: Table 3 instead of Table 3.3. The same for Table 5 in Page 4 line 367.
Reply: thank you, a macro of word generates this error.
- Page 14, line 378: “According to [54] and [55], […]” instead of “According to 54 and 55, […]”
Reply: suggestion has been followed
- Page 14, line 379: Figure 9 instead of Fig9.
Reply: suggestion has been followed
- In the FPGA implementation of the control, floating point representation has been chosen instead of the fixed-point representation. Could the author comment on this choice? With FP the precision is improved but the required resources are higher.
Reply: The use of a FPGA allows managing the execution-time of the control algorithm and it is possible to execute more mathematical operation in the same time thanks to low-level implementation. Moreover, by implementing basic mathematical operations (product, sum and subtraction) in FP at 32bit single-precision the same FPGA resources required can be reused during the execution of the control algorithm. In this way, it is possible to reduce the overall FPGA resources and optimize the overall control algorithm. For this reason I chose the FP resolution for the numerical variables. It is interesting to note that overall control algorithm required less 50% of the FPGA resources.
- Concerning the comparisons presented in Figure 42-43-44-45, have they been achieved by using different filter parameters in the experimental setup?
Reply: Sentence has been changed into:
“Figures 42-44 show the comparison between the simulated (blue bars) and the experimental (yellow bars) THD% values obtained with Sinusoidal (Figure 42), THI (Figure 43) and SFO (Figure 44) as reference signals for each modulation techniques taken into account with the designed filter discussed in 2.2.”
Again thank you for your time and help
Round 2
Reviewer 1 Report
N/A
Author Response
I want to thank the reviewer for his time devoted to the study of this long paper.
Reviewer 2 Report
The paper can be accepted in the present form.
Author Response

(The authors gave the same response as above.)

Reviewer 3 Report
The author replied to all the comments and the paper has been improved.
According to the reviewer, the author could explicitly mention that both the modulation technique and the LCL filter concur to define the harmonic content presented in the comparison of the modulation techniques. For example after table 8 when he claims: “It is interesting to note that the SPD and SFOPD present the lower values of the PTHD% as regard to the currents"
Page 17, Line 458: “[…] it is interesting […]” instead of “[…] it interesting […]”
Author Response
Dear Reviewer,
I want to thank you for the suggestions and the time devoted in this review.
All required changes have been made.
Best regards,
Fabio Viola